# 🔥 RELBENCH: A Benchmark for Deep Learning on Relational Databases

Joshua Robinson[1]*, Rishabh Ranjan[1]*, Weihua Hu[2]*, Kexin Huang[1]*,
Jiaqi Han[1], Alejandro Dobles[1], Matthias Fey[2], Jan E. Lenssen[2,3],
Yiwen Yuan[2], Zecheng Zhang[2], Xinwei He[2], Jure Leskovec[1,2]
[1]Stanford University [2]Kumo.AI [3]Max Planck Institute for Informatics

https://relbench.stanford.edu

## Abstract

We present RELBENCH, a public benchmark for solving predictive tasks over relational databases with graph neural networks. RELBENCH provides databases and tasks spanning diverse domains and scales, and is intended to be a foundational infrastructure for future research. We use RELBENCH to conduct the first comprehensive study of Relational Deep Learning (RDL) (Fey *et al.*, 2024), which combines graph neural network predictive models with (deep) tabular models that extract initial entity-level representations from raw tables. End-to-end learned RDL models fully exploit the predictive signal encoded in primary-foreign key links, marking a significant shift away from the dominant paradigm of manual feature engineering combined with tabular models. To thoroughly evaluate RDL against this prior gold-standard, we conduct an in-depth user study where an experienced data scientist manually engineers features for each task. In this study, RDL learns better models whilst reducing human work needed by more than an order of magnitude. This demonstrates the power of deep learning for solving predictive tasks over relational databases, opening up many new research opportunities enabled by RELBENCH.

## 1 Introduction

Relational databases are the most widely used database management system, underpinning much of the digital economy. Their popularity stems from their table storage structure, making maintenance relatively easy, and data simple to access using powerful query languages such as SQL. Because of their popularity, AI systems across a wide variety of domains are built using data stored in relational databases, including e-commerce, social media, banking systems, healthcare, manufacturing, and open-source scientific repositories (Johnson *et al.*, 2016; PubMed, 1996).

Despite the importance of relational databases, the rich relational information is typically foregone, as no model architecture is capable of handling varied database structures. Instead, data is "flattened" into a simpler format such as a single table, often by manual feature engineering, on which standard tabular models can be used (Kaggle, 2022). This results in a significant loss in predictive signal, and creates a need for data extraction pipelines that frequently cause bugs and add to software complexity.

To fully exploit the predictive signal encoded in the relations between entities, a new proposal is to re-cast relational data as an exact *graph* representation, with a node for each entity in the database, edges indicating primary-foreign key links, and node features extracted using deep tabular models, an approach termed *Relational Deep Learning* (RDL) (Fey *et al.*, 2024). The graph representation allows Graph Neural Networks (GNNs) (Gilmer *et al.*, 2017; Hamilton *et al.*, 2017) to be used as predictive models. RDL is the first approach for an end-to-end learnable neural network model with access to all possible predictive signal in a relational databases, and has the potential to unlock new

---

*Equal contribution, order chosen randomly. First authors may swap the ordering for professional purposes.

38th Conference on Neural Information Processing Systems (NeurIPS 2024) Track on Datasets and Benchmarks.

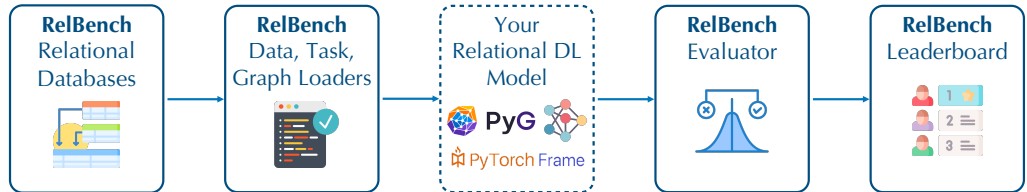

Figure 1: **RELBENCH** enables training and evaluation of deep learning models on relational databases. RELBENCH supports framework agnostic data loading, task specification, standardized data splitting, standardized evaluation metrics, and a leaderboard for tracking progress. RELBENCH also includes a pilot implementation of the relational deep learning blueprint of Fey *et al.* (2024).

levels of predictive power. However, the development of relational deep learning is limited by a complete lack of infrastructure to support research, including: (i) standardized benchmark databases and tasks to compare methods, (ii) initial implementation of RDL, including converting data to graph form and GNN training, and (iii) a pilot study of the effectiveness of relational deep learning.

Here we present RELBENCH, the first benchmark for relational deep learning. RELBENCH is intended to be the foundational infrastructure for future research into relational deep learning, providing a comprehensive set of databases across a variety of domains, including *e-commerce*, *Q&A platforms*, *medical*, and *sports* databases. RELBENCH databases span orders of magnitude in size, from 74K entities to 41M entities, and have very different time spans, between 2 weeks and 55 years of training data. They also vary significantly in their relational structure, with the total number of tables varying between 3 and 15, and total number of columns varying from 15 to 140. Each database comes with multiple predictive tasks, 30 in total, including entity classification/regression and recommendation tasks, each chosen for their real-world significance.

In addition to databases and tasks, we release open-source software designed to make relational deep learning widely available. This includes (i) the RELBENCH Python package for easy database and task loading, (ii) the first open-source implementation of relational deep learning, designed to be easily modified by researchers, and (iii) a public leaderboard for tracking progress. We comprehensively benchmark our initial RDL implementation on all RELBENCH tasks, comparing to various baselines.

The most important baseline we compare to is a strong "data scientist" approach, for which we recruited an experienced individual to solve each task by manually engineering features and feeding them into tabular models. This approach is the current gold-standard for building predictive models on relational databases. The study, which we open source for reproducibility, finds that RDL models match or outperform the data scientist's models in accuracy, whilst reducing human hours worked by $96\%$, and lines of code by $94\%$ on average. This constitutes the first empirical demonstration of the central promise of RDL, and points to a long-awaited end-to-end deep learning solution for relational data.

Our website[2] is a comprehensive entry point to RDL, describing RELBENCH databases and tasks, access to code on GitHub, the full relational deep learning blueprint, and tutorials for adding new databases and tasks to RELBENCH to allow researchers to experiment with their problems of interest.

## 2   Overview and Design

RELBENCH provides a collection of diverse real-world **relational databases** along with a set of realistic **predictive tasks** associated with each database. Concretely, we provide:

- **Relational databases**, consisting of a set of tables connected via primary-foreign key relationships. Each table has columns storing diverse information about each entity. Some tables also come with *time columns*, indicating the time at which the entity is created (*e.g.*, transaction date).
- **Predictive tasks over a relational database**, which are defined by a **training table** (Fey *et al.*, 2024) with columns for Entity ID, seed time, and target labels.The seed time indicates *at which time* the target is to be predicted, filtering future data.

Next we outline key design principles of RELBENCH with an emphasis on data curation, data splits, research flexibility, and open-source implementation.

---

[2]https://relbench.stanford.edu.

Table 1: **Statistics of RELBENCH datasets.** Datasets vary significantly in the number of tables, total number of rows, and number of columns. In this table, we only count rows available for test inference, i.e., rows upto the test time cutoff.

| Name | Domain | #Tasks | Tables | | | Timestamp (year-mon-day) | | |
|---|---|---|---|---|---|---|---|---|
| | | | #Tables | #Rows | #Cols | Start | Val | Test |
| rel-amazon | E-commerce | 7 | 3 | 15,000,713 | 15 | 2008-01-01 | 2015-10-01 | 2016-01-01 |
| rel-avito | E-commerce | 4 | 8 | 20,679,117 | 42 | 2015-04-25 | 2015-05-08 | 2015-05-14 |
| rel-event | Social | 3 | 5 | 41,328,337 | 128 | 1912-01-01 | 2012-11-21 | 2012-11-29 |
| rel-f1 | Sports | 3 | 9 | 74,063 | 67 | 1950-05-13 | 2005-01-01 | 2010-01-01 |
| rel-hm | E-commerce | 3 | 3 | 16,664,809 | 37 | 2019-09-07 | 2020-09-07 | 2020-09-14 |
| rel-stack | Social | 5 | 7 | 4,247,264 | 52 | 2009-02-02 | 2020-10-01 | 2021-01-01 |
| rel-trial | Medical | 5 | 15 | 5,434,924 | 140 | 2000-01-01 | 2020-01-01 | 2021-01-01 |
| Total | | 30 | 51 | 103,466,370 | 489 | / | / | / |

**Data Curation**. Relational databases are widespread, so there are many candidate predictive tasks. For the purpose of benchmarking we carefully curate a collection of relational databases and tasks chosen for their rich relational structure and column features. We also adopt the following principles:

- **Diverse domains:** To ensure algorithms developed on RELBENCH will be useful across a wide range of application domains, we select real-world relational databases from diverse domains.
- **Diverse task types:** Tasks cover a wide range of real-world use-cases, including three representative task types: entity classification, entity regression, and recommendation.

RELBENCH databases are summarized in Table 1, covering E-commerce, social, medical, and sports domains. The databases vary significantly in the numbers of rows (*i.e.*, data scale) the number of columns and tables, as well as the time ranges of the databases. Tasks are summarized in Table 2, each corresponding to a predictive problem of practical interest such as predicting customer churn, predicting the number of adverse events in a clinical trial, and recommending posts to users.

**Data Splits**. Data is split temporally, with models trained on rows up to VAL_TIMESTAMP, validated on the rows between VAL_TIMESTAMP and TEST_TIMESTAMP, and tested on the rows after TEST_TIMESTAMP. Our implementation carefully hides data after TEST_TIMESTAMP during inference to systematically avoid test time data leakage (Kapoor and Narayanan, 2023), and uses an elegant solution proposed by Fey *et al.* (2024) to avoid time leakage during training and validation through temporal neighbor sampling. In general, it is the designers responsibility to avoid time leakage. We recommend using our carefully tested implementation where possible.

**Research Flexibility**. RELBENCH is designed to allow significant freedom in future research directions. For example, RELBENCH tasks share the same (VAL_TIMESTAMP and TEST_TIMESTAMP) splits across tasks within the same relational database. This opens up exciting opportunities for multi-task learning and pre-training to simultaneously improve different predictive tasks within the same relational database. We also expose the logic for converting databases into graphs. This allows future work to consider modified graph constructions, or creative uses of the raw data.

**Open-source RDL Implementation**. As well as datasets and tasks, we provide the first open-source implementation of relational deep learning. See Figure 2 of Fey *et al.* (2024) for a high-level overview. A neural network is learned over a heterogeneous temporal graph that exactly represents the database in order to make prediction over nodes (for entity classification and regression) and links (for recommendation). Our implementation is built on top of PyTorch Frame (Hu *et al.*, 2024) for extracting initial node embeddings from raw table features, and PyTorch Geometric (Fey and Lenssen, 2019) for GNN modeling. See Section A for details.

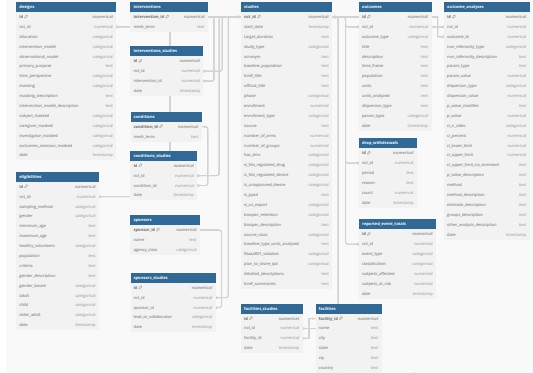

Figure 2: Example RELBENCH schema for rel-trial database. RELBENCH databases have complex relational structure and rich column features.

Table 2: Full list of predictive tasks for each RELBENCH dataset (introduced in Table 1).

| Dataset | Task name | Task type | #Rows of training table | | | #Unique Entities | %train/test Entity Overlap | #Dst Entities |
|---------|-----------|-----------|-------|------------|------|------------------|---------------------------|---------------|
| | | | Train | Validation | Test | | | |
| rel-amazon | user-churn | entity-cls | 4,732,555 | 409,792 | 351,885 | 1,585,983 | 88.0 | — |
| | item-churn | entity-cls | 2,559,264 | 177,689 | 166,842 | 416,352 | 93.1 | — |
| | user-ltv | entity-reg | 4,732,555 | 409,792 | 351,885 | 1,585,983 | 88.0 | — |
| | item-ltv | entity-reg | 2,707,679 | 166,978 | 178,334 | 427,537 | 93.5 | — |
| | user-item-purchase | recommendation | 5,112,803 | 351,876 | 393,985 | 1,632,909 | 87.4 | 12,562,384 |
| | user-item-rate | recommendation | 3,667,157 | 257,939 | 292,609 | 1,481,360 | 81.0 | 7,665,611 |
| | user-item-review | recommendation | 2,324,177 | 116,970 | 127,021 | 894,136 | 74.1 | 5,406,835 |
| rel-avito | ad-ctr | entity-reg | 5,100 | 1,766 | 1,816 | 4,997 | 59.8 | — |
| | user-clicks | entity-cls | 59,454 | 21,183 | 47,996 | 66,449 | 45.3 | — |
| | user-visits | entity-cls | 86,619 | 29,979 | 36,129 | 63,405 | 64.6 | — |
| | user-ad-visit | recommendation | 86,616 | 29,979 | 36,129 | 63,402 | 64.6 | 3,616,174 |
| rel-event | user-attendance | entity-reg | 19,261 | 2,014 | 2,006 | 9,694 | 14.6 | — |
| | user-repeat | entity-cls | 3,842 | 268 | 246 | 1,514 | 11.5 | — |
| | user-ignore | entity-cls | 19,239 | 4,185 | 4,010 | 9,799 | 21.1 | — |
| rel-f1 | driver-dnf | entity-cls | 11,411 | 566 | 702 | 821 | 50.0 | — |
| | driver-top3 | entity-cls | 1,353 | 588 | 726 | 134 | 50.0 | — |
| | driver-position | entity-reg | 7,453 | 499 | 760 | 826 | 44.6 | — |
| rel-hm | user-churn | entity-cls | 3,871,410 | 76,556 | 74,575 | 1,002,984 | 89.7 | — |
| | item-sales | entity-reg | 5,488,184 | 105,542 | 105,542 | 105,542 | 100.0 | — |
| | user-item-purchase | recommendation | 3,878,451 | 74,575 | 67,144 | 1,004,046 | 89.2 | 13,428,473 |
| rel-stack | user-engagement | entity-cls | 1,360,850 | 85,838 | 88,137 | 88,137 | 97.4 | — |
| | user-badge | entity-cls | 3,386,276 | 247,398 | 255,360 | 255,360 | 96.9 | — |
| | post-votes | entity-reg | 2,453,921 | 156,216 | 160,903 | 160,903 | 97.1 | — |
| | user-post-comment | recommendation | 21,239 | 825 | 758 | 11,453 | 59.9 | 44,940 |
| | post-post-related | recommendation | 5,855 | 226 | 258 | 5,924 | 8.5 | 7,456 |
| rel-trial | study-outcome | entity-cls | 11,994 | 960 | 825 | 13,779 | 0.0 | — |
| | study-adverse | entity-reg | 43,335 | 3,596 | 3,098 | 50,029 | 0.0 | — |
| | site-success | entity-reg | 151,407 | 19,740 | 22,617 | 129,542 | 42.0 | — |
| | condition-sponsor-run | recommendation | 36,934 | 2,081 | 2,057 | 3,956 | 98.4 | 533,624 |
| | site-sponsor-run | recommendation | 669,310 | 37,003 | 27,428 | 445,513 | 48.3 | 1,565,463 |

# 3 RELBENCH Datasets

RELBENCH contains 7 datasets each
with rich relational structure, providing a challenging environment for developing and comparing relational deep learning methods (see Figure 2 for an example). The datasets are carefully processed from real-world relational databases and span diverse domains and sizes. Each database is associated with multiple individual predictive tasks defined in Section 4. Detailed statistics of each dataset can be found in Table 1. We briefly describe each dataset.

**rel-amazon**. The Amazon E-commerce database records products, users, and reviews across Amazon's E-commerce platform. It contains rich information about products and reviews. Products include the price and category of each, reviews have the overall rating, whether the user has actually bought the product, and the text of the review itself. We use the subset of book-related products.

**rel-f1**. The F1 database tracks all-time Formula 1 racing data and statistics since 1950. It provides detailed information for various stakeholders including drivers, constructors, engine manufacturers, and tyre manufacturers. Highlights include data on all circuits (*e.g.* geographical details), and full historical data from every season. This includes overall standings, race results, and more specific data like practice sessions, qualifying positions, sprints, and pit stops.

**rel-stack**. Stack Exchange is a network of question-and-answer websites on different topics, where questions, answers, and users are subject to a reputation award process. The reputation system allows the sites to be self-moderating. The database includes detailed records of activity including user biographies, posts and comments (with raw text), edit histories, voting, and related posts. In our benchmark, we use the stats-exchange site.

**rel-trial**. The clinical trial database is curated from AACT initiative, which consolidates all protocol and results data from studies registered on ClinicalTrials.gov. It offers extensive information about clinical trials, including study designs, participant demographics, intervention details, and outcomes. It is an important resource for health research, policy making, and therapeutic development.

**rel-hm**. The H&M relational database hosts extensive customer and product data for online shopping experiences across its extensive network of brands and stores. This database includes

detailed customer purchase histories and a rich set of metadata, encompassing everything from basic demographic information to extensive details about each product available.

**rel-event**. The Event Recommendation database is obtained from user data on a mobile app called Hangtime. This app allows users to keep track of their friends' social plans. The database contains data on user actions, event metadata, and demographic information, as well as users' social relations, which captures how social relations can affect user behavior. Data is fully anonymized, with no personally identifiable information (such as names or aliases) available.

**rel-avito**. Avito is a leading online advertisement platform, providing a marketplace for users to buy and sell a wide variety of products and services, including real estate, vehicles, jobs, and goods. The Avito Context Ad Clicks dataset on Kaggle is part of a competition aimed at predicting whether an ad will be clicked based on contextual information. This dataset includes user searches, ad attributes, and other related data to help build predictive models.

**Data Provenance**. All data is sourced from publicly available repositories with licenses permitting usage for research purposes. See Appendix E for details of data sources, licenses, and more.

## 4  Predictive Tasks on RELBENCH Datasets

RELBENCH introduces 30 new predictive tasks defined over the databases introduced in Section 2. A full list of tasks is given in Table 2, with high-level descriptions given in Appendix B (and our website) due to space limitations. Tasks are grouped into three task types: entity classification (Section 4.1), entity regression (Section 4.2), and entity link prediction (Section 4.3). Tasks differ significantly in the number of train/val/test entities, number of unique entities (the same entity may appear multiple times at different timestamps), and the proportion of test entities seen during training. Note this is not data leakage, since entity predictions are timestamp dependent, and can change over time. Tasks with no overlap are pure inductive tasks, whilst other tasks are (partially) transductive.

Table 3: Entity classification results (AUROC, higher is better) on RELBENCH. Best values are in bold. See Table 6 in Appendix C for standard deviations.

| Dataset | Task | Split | LightGBM | RDL | Rel. Gain of RDL |
|---|---|---|---|---|---|
| rel-amazon | user-churn | Val | 52.05 | **70.45** | 35.35 % |
| | | Test | 52.22 | **70.42** | 34.86 % |
| | item-churn | Val | 62.39 | **82.39** | 32.06 % |
| | | Test | 62.54 | **82.81** | 32.40 % |
| rel-avito | user-visits | Val | 53.31 | **69.65** | 30.66 % |
| | | Test | 53.05 | **66.20** | 24.78 % |
| | user-clicks | Val | 55.63 | **64.73** | 16.35 % |
| | | Test | 53.60 | **65.90** | 22.96 % |
| rel-event | user-repeat | Val | 67.76 | **71.25** | 5.15 % |
| | | Test | 68.04 | **76.89** | 13.02 % |
| | user-ignore | Val | 87.96 | **91.70** | 4.25 % |
| | | Test | 79.93 | **81.62** | 2.12 % |
| rel-f1 | driver-dnf | Val | 68.42 | **71.36** | 4.31 % |
| | | Test | 68.56 | **72.62** | 5.93 % |
| | driver-top3 | Val | 67.76 | **77.64** | 14.57 % |
| | | Test | 73.92 | **75.54** | 2.20 % |
| rel-hm | user-churn | Val | 56.05 | **70.42** | 25.63 % |
| | | Test | 55.21 | **69.88** | 26.59 % |
| rel-stack | user-engagement | Val | 65.12 | **90.21** | 38.53 % |
| | | Test | 63.39 | **90.59** | 42.91 % |
| | user-badge | Val | 65.39 | **89.86** | 37.43 % |
| | | Test | 63.43 | **88.86** | 40.08 % |
| rel-trial | study-outcome | Val | **68.30** | 68.18 | −0.19 % |
| | | Test | **70.09** | 68.60 | −2.13 % |
| | Average | Val | 64.18 | **76.49** | 20.34 % |
| | | Test | 63.66 | **75.83** | 20.48 % |

### 4.1  Entity Classification

The first task type is entity-level classification. The task is to predict binary labels of a given entity at a given seed time. We use the ROC-AUC (Hanley and McNeil, 1983) metric for evaluation (higher is better). We compare to a LightGBM classifier baseline over the raw entity table features. Note that here only information from the single entity table is used.

**Experimental results.**  Results are given in Table 3, with RDL outperforming or matching baselines in all cases. Notably, LightGBM achieves similar performance to RDL on the study-outcome task from rel-trial. This task has extremely rich features in the target table (28 columns total), giving the LightGBM many potentially useful features even without feature engineering. It is an interesting research question how to design RDL models better able to extract these features and unify them with cross-table information in order to outperform the LightGBM model on this dataset.

Table 4: Entity regression results (MAE, lower is better) on RELBENCH. Best values are in bold. See Table 7 in Appendix C for standard deviations.

| Dataset | Task | Split | Global Zero | Global Mean | Global Median | Entity Mean | Entity Median | LightGBM | RDL | Rel. Gain of RDL |
|---------|------|-------|-------------|-------------|---------------|-------------|---------------|----------|-----|------------------|
| rel-amazon | user-ltv | Val | 14.141 | 20.740 | 14.141 | 17.685 | 15.978 | 14.141 | **12.132** | 14.21 % |
| | | Test | 16.783 | 22.121 | 16.783 | 19.055 | 17.423 | 16.783 | **14.313** | 14.72 % |
| | item-ltv | Val | 72.096 | 78.110 | 59.471 | 80.466 | 68.922 | 55.741 | **45.140** | 19.02 % |
| | | Test | 77.126 | 81.852 | 64.234 | 78.423 | 66.436 | 60.569 | **50.053** | 17.36 % |
| rel-avito | ad-ctr | Val | 0.048 | 0.048 | 0.040 | 0.044 | 0.044 | 0.037 | **0.037** | 2.21 % |
| | | Test | 0.052 | 0.051 | 0.043 | 0.046 | 0.046 | **0.041** | 0.041 | −0.18 % |
| rel-event | user-attendance | Val | 0.262 | 0.457 | 0.262 | 0.296 | 0.268 | 0.262 | **0.255** | 2.65 % |
| | | Test | 0.264 | 0.470 | 0.264 | 0.304 | 0.269 | 0.264 | **0.258** | 1.97 % |
| rel-f1 | driver-position | Val | 11.083 | 4.334 | 4.136 | 7.181 | 7.114 | 3.450 | **3.193** | 7.44 % |
| | | Test | 11.926 | 4.513 | 4.399 | 8.501 | 8.519 | 4.170 | **4.022** | 3.56 % |
| rel-hm | item-sales | Val | 0.086 | 0.142 | 0.086 | 0.117 | 0.086 | 0.086 | **0.065** | 24.50 % |
| | | Test | 0.076 | 0.134 | 0.076 | 0.111 | 0.078 | 0.076 | **0.056** | 26.90 % |
| rel-stack | post-votes | Val | 0.062 | 0.146 | 0.062 | 0.102 | 0.064 | 0.062 | **0.059** | 4.19 % |
| | | Test | 0.068 | 0.149 | 0.068 | 0.106 | 0.069 | 0.068 | **0.065** | 4.11 % |
| rel-trial | study-adverse | Val | 57.083 | 75.008 | 56.786 | 57.083 | 57.083 | **45.774** | 46.290 | −1.13 % |
| | | Test | 57.930 | 73.781 | 57.533 | 57.930 | 57.930 | **44.011** | 44.473 | −1.05 % |
| | site-success | Val | 0.475 | 0.462 | 0.475 | 0.447 | 0.450 | 0.417 | **0.401** | 3.87 % |
| | | Test | 0.462 | 0.468 | 0.462 | 0.448 | 0.441 | 0.425 | **0.400** | 5.86 % |
| | Average | Val | 17.260 | 19.939 | 15.051 | 18.158 | 16.668 | 13.330 | **11.952** | 8.55 % |
| | | Test | 18.299 | 20.393 | 15.985 | 18.325 | 16.801 | 14.045 | **12.631** | 8.14 % |

## 4.2 Entity Regression

Entity-level regression tasks involve predicting numerical labels of an entity at a given seed time. We use Mean Absolute Error (MAE) as our metric (lower is better). We consider the following baselines:

- **Entity mean/median** calculates the mean/median label value for each entity in training data and predicts the mean/median value for the entity.
- **Global mean/median** calculates the global mean/median label value over the training data and predicts the same mean/median value across all entities.
- **Global zero** predicts zero for all entities.
- **LightGBM** learns a LightGBM (Ke *et al.*, 2017) regressor over the raw entity features to predict the numerical targets. Note that only information from the single entity table is used.

**Experimental results**. Results in Table 4 show our RDL implementation outperforms or matches baselines in all cases. A number of tasks, such as driver-position and study-adverse, have matching performance up to statistical significance, suggesting some room for improvement. We analyze this further in Appendix D, identifying one potential cause, suggesting an opportunity for improved performance for regression tasks.

## 4.3 Recommendation

Finally, we also introduce recommendation tasks on pairs of entities. The task is to predict a list of top $K$ target entities given a source entity at a given seed time. The metric we use is Mean Average Precision (MAP) @$K$, where $K$ is set per task (higher is better). We consider the following baselines:

- **Global popularity** computes the top $K$ most popular target entities (by count) across the entire training table and predict the $K$ globally popular target entities across all source entities.
- **Past visit** computes the top $K$ most visited target entities for each source entity within the training table and predict those past-visited target entities for each entity.
- **LightGBM** learns a LightGBM (Ke *et al.*, 2017) classifier over the raw features of the source and target entities (concatenated) to predict the link. Additionally, global popularity and past visit ranks are also provided as inputs.

For recommendation, it is also important to ensure a certain density of links in the training data in order for there to be sufficient predictive signal. In Appendix B we report statistics on the average number of destination entities each source entity links to. For most tasks the density is $\geq 1$, with the exception of rel-stack which is more sparse, but is included to test in extreme sparse settings.

Table 5: Recommendation results (MAP, higher is better) on RELBENCH. Best values are in bold. See Table 8 in Appendix C for standard deviations.

| Dataset | Task | Split | Global Popularity | Past Visit | LightGBM | RDL (GraphSAGE) | RDL (ID-GNN) | Rel. Gain of RDL |
|---------|------|-------|-------------------|------------|----------|------------------|---------------|-------------------|
| rel-amazon | user-item-purchase | Val | 0.31 | 0.07 | 0.18 | **1.53** | 0.13 | 397.55 % |
| | | Test | 0.24 | 0.06 | 0.16 | **0.74** | 0.10 | 204.74 % |
| | user-item-rate | Val | 0.16 | 0.09 | 0.22 | **1.42** | 0.15 | 550.12 % |
| | | Test | 0.15 | 0.07 | 0.17 | **0.87** | 0.12 | 395.92 % |
| | user-item-review | Val | 0.18 | 0.05 | 0.14 | **1.03** | 0.11 | 476.06 % |
| | | Test | 0.11 | 0.04 | 0.09 | **0.47** | 0.09 | 313.07 % |
| rel-avito | user-ad-visit | Val | 0.01 | 3.66 | 0.17 | 0.09 | **5.40** | 47.37 % |
| | | Test | 0.00 | 1.95 | 0.06 | 0.02 | **3.66** | 87.09 % |
| rel-hm | user-item-purchase | Val | 0.36 | 1.07 | 0.44 | 0.92 | **2.64** | 145.60 % |
| | | Test | 0.30 | 0.89 | 0.38 | 0.80 | **2.81** | 214.49 % |
| rel-stack | user-post-comment | Val | 0.03 | 2.05 | 0.04 | 0.43 | **15.17** | 640.05 % |
| | | Test | 0.02 | 1.42 | 0.04 | 0.11 | **12.72** | 795.15 % |
| | post-post-related | Val | 0.47 | 0.00 | 1.62 | 0.00 | **7.76** | 378.26 % |
| | | Test | 1.46 | 1.74 | 2.00 | 0.07 | **10.83** | 440.27 % |
| rel-trial | condition-sponsor-run | Val | 2.63 | 8.58 | 4.88 | 3.12 | **11.33** | 32.05 % |
| | | Test | 2.52 | 8.42 | 4.82 | 2.89 | **11.36** | 34.89 % |
| | site-sponsor-run | Val | 4.91 | 15.90 | 10.92 | 14.09 | **17.43** | 9.65 % |
| | | Test | 3.75 | 17.31 | 8.40 | 10.70 | **19.00** | 9.74 % |
| | Average | Val | 1.01 | 3.50 | 2.07 | 2.51 | **6.68** | 297.41 % |
| | | Test | 0.95 | 3.55 | 1.79 | 1.85 | **6.74** | 277.26 % |

**Experimental results**. Results are given in Table 5. We find that either the RDL implementation using GraphSAGE (Hamilton *et al.*, 2017), or ID-GNN (You *et al.*, 2021) as the GNN component performs best, often by a very significant margin. ID-GNN excels in cases were predictions are entity-specific (*i.e.*, Past Visit baseline outperforms Global Popularity), whilst the plain GNN excels in the reverse case. This reflects the inductive biases of each model, with GraphSAGE being able to learn structural features, and ID-GNN able to take into account the specific node ID.

## 5 Expert Data Scientist User Study

To test RDL in the most challenging circumstances possible, we undertake a human trial wherein a data scientist solves each task by manually designing features and feeds them into tabular methods such at LightGBM or XGBoost (Chen and Guestrin, 2016; Ke *et al.*, 2017). This represents the prior gold-standard for building predictive models on relational databases (Heaton, 2016), and the key point of comparison for RDL.

We structure our user study along the five main data science workflow steps:

1. **Exploratory data analysis (EDA):** Explore the dataset and task to understand its characteristics, including what column features there are, and if there is any missing data.
2. **Feature ideation:** Based on EDA and intuition from prior experiences, propose a set of entity-level features that the data scientist believes may contain predictive signal for the task.
3. **Feature enginnering:** Using query languages such as SQL to compute the proposed features, and add them as extra columns to the target table of interest.
4. **Tabular ML:** Run tabular methods such as LightGBM or XGBoost on the table with extra features to produce a predictive model, and record the test performance.
5. **Post-hoc analysis of feature importance (Optional):** Common tools include SHAP and LIME, which aim to explain the contribution of each input feature to the final performance.

Consider for example the rel-hm dataset (schema in Appendix E) and the task of predicting customer churn. Here the CUSTOMER table only contains simple biographical information such as username and joining date. To capture more predictive information, additional features, such as *time since last purchase*, can be computed using the other tables, and added to the CUSTOMER table. We give a detailed walk-through of the data scientist's work process for solving this specific task in Appendix D. We strongly encourage the interested reader to review this, as it highlights the significant amount of task-specific effort that this workflow necessitates.

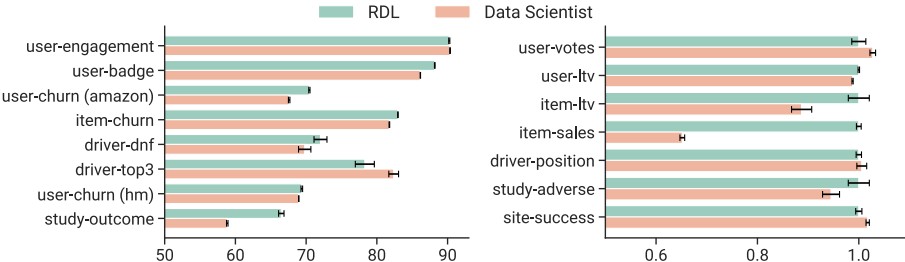

Figure 3: **RDL vs. Data Scientist.** Relational Deep Learning matches or outperforms the data scientist in 11 of 15 tasks. Left shows entity classification AUROC, right shows entity regression, reporting MAE normalized so that the RDL MAE is always 1.

**Limitations of Manual Feature Engineering**. This workflow suffers from several fundamental limitations. Most obviously, since features are hand designed they only capture part of the predictive signal in the database, useful signal is easily missed. Additionally, feature complexity is limited by human reasoning abilities, meaning that higher-order interactions between entities are often overlooked. Beyond predictive signal, the other crucial limitation of feature engineering is its extremely manual nature—every time a new model is built a data scientist has to repeat this process, requiring many hours of human labor, and significant quantities of new SQL code to design features (Zheng and Casari, 2018). Our RDL models avoid these limitations (see Section 5).

**Data Scientist**. To conduct a thorough comparison to this process, we recruit a high-end data scientist with Stanford CS MSc degree, 4.0 GPA, and 5 years of experience of building machine learning models in the financial industry. This experience includes a significant amount of time building machine learning models in exactly above five steps, as well as broader data science expertise.

**User Study Protocol**.

Because of the open-ended nature of feature engineering and model development, we follow a specific protocol for the user study in order to standardize the amount of effort dedicated to each dataset and task. Tracking the 5 steps outlined above, we impose the following rules:

1. **EDA:** The time allotted for data exploration is capped at 4 hours. This threshold was chosen to give the data scientist enough time to familiarize themselves with the schema, visualize key relationships and distributions, and take stock of any outliers in the dataset, while providing a reasonable limit to the effort applied.

2. **Feature ideation:** Feature ideation is performed manually with pen and paper, and is limited to 1 hour. In practice, the data scientist found that 1 hour was plenty of time to enumerate all promising features at that time, especially since many ideas naturally arise during the EDA process already.

3. **Feature engineering:** The features described during the ideation phase are then computed using SQL queries. The time taken to write SQL code to generate the features is unconstrained in order to eliminate code writing speed as a factor in the study. We do, however, record code writing time for our timing benchmarking. This stage presented the most variability in terms of time commitment, partly because it is unconstrained, but mostly because the implementation complexity of the features itself is highly variable.

4. **Tabular ML:** For tabular ML training, we provide a standardized LightGBM training script including comprehensive hyperparameter tuning. The data scientist needs only to feed the table full of engineered features into this training script, which returns test performance results. However, there is some non-trivial amount of work required to transform the output of the SQL queries from the previous section into the Python objects (arrays) required for training LightGBM. Again, the time taken for this additional pre-preocessing is recorded.

5. **Post-hoc analysis of feature importance:** Finally, after successfully training a model, an evaluation of model predictions and feature importance is carried out. This mostly serves as a general sanity check and an interesting corollary of the data scientist's work that provides task-specific insights (see Appendix D). In practice, this took no more than a few minutes per task and this time was not counted toward the total time commitment.

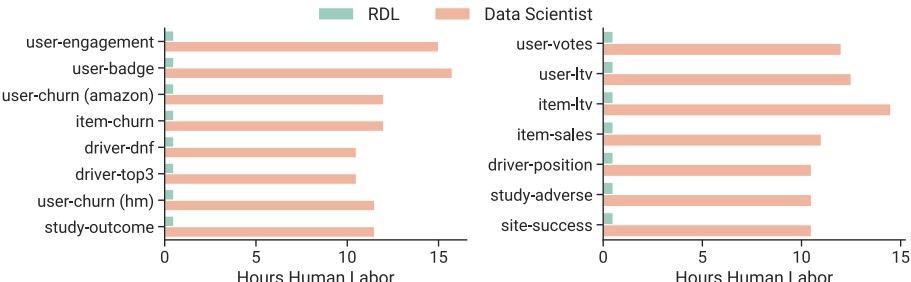

Figure 4: **RDL vs. Data Scientist.** Relational Deep Learning reduces the hours of human work required to solve a new task by 96% on average (from 12.3 to 0.5 hours). Left shows node-level classification, right shows node-level regression.

**Reproducibility**. All of the data scientist's workings are released[3] to ensure reproducibility and demonstrate the significant lengths gone through to build as accurate models as possible. In Appendix D we walk through a complete example for a single dataset and task, showing the data-centric insights it yields. An important by-product is a close analysis of which features contribute to model performance, which we believe will help inspire future well-motivated RDL research directions.

**Results**. As well as (i) raw predictive power, we compare the data scientist to our RDL models in terms of (ii) hours of human work, and (iii) number of new lines of code required to solve each task. We measure the *marginal* effort, meaning that we do not include code infrastructure that is reused across tasks, including for example data loading logic and training scripts for RDL or LightGBM models. Accordingly, we only compare model development, not data preparation/loading. Indeed the data loading pipeline is shared between RDL and the data scientist, so RDL does not introduce any significant overheads for data loading/preparation over a data scientist's approach. We believe that accelerating model development (apart from data loading) is valuable in many use cases where engineers need to solve many different predictive tasks over a single database.

**Summary**. Figures 3, 4, and 5 show that RDL learns highly predictive models, outperforming the data scientist in 11 of 15 tasks, whilst reducing hours worked by 96% on average, and lines of code by 94% on average. On average, it took the data scientist 12.3 hours to solve each task using traditional feature engineering. By contrast it takes roughly 30 minutes to solve a task with RDL. This observation is *the* central value proposition of relational deep learning, pointing the way to unlocking new levels of predictive power, and potentially a new economic model for solving predictive tasks on relational databases. Replacing hand-crafted solutions with end-to-end learnable models has been a key takeaway from the last 15 years of AI research. It is therefore remarkable how little impact deep learning has had on ML on relational databases, one of the most widespread applied ML use cases. To the best of our knowledge, is RDL the first deep learning approach for relational databases that has demonstrated efficacy compared with established data science workflows. We highlight that all RELBENCH tasks were solved with a single set of default hyperparameters (with 2 exceptions requiring small modifications to learning rate, number of epochs, and GNN aggregation function). This demonstrates the robustness of RDL, and that the performance of RDL in Figure 3 is not due to extensive hyperparamter search. Indeed, the single set of RDL hyperparameters is compared to a carefully tuned LightGBM, which was allowed to search over 10 sets of hyperparameters.

**Predictive Power**. Results shown in Figures 3. Whilst outperforming the data scientist in 11 of 15 tasks, we note that RDL best outperforms the data scientist on classification tasks, struggling more on regression. Indeed it was necessary for us to apply a "boosting" to the RDL model to improve performance (see Appendix D). Even with boosting, the data scientist model outperforms RDL in several cases. One cause we identify is that the MLP output head of the GNN is poorly suited to regression tasks (see Appendix D). This suggests an opportunity for improved output heads for regression tasks. We stress that our RDL implementation is an *initial* demonstration. We believe there is significant scope for new research leading to large improvements in performance. In particular, ideas from graph ML, deep tabular ML, and time-series modeling are well suited to advance RDL.

**Human Work**. Results shown in Figure 4. In our user study RDL required 96% less hours work to solve a new task, compared to the data scientist work flow. The RDL solutions always took less

---

[3]See https://github.com/snap-stanford/relbench-user-study.

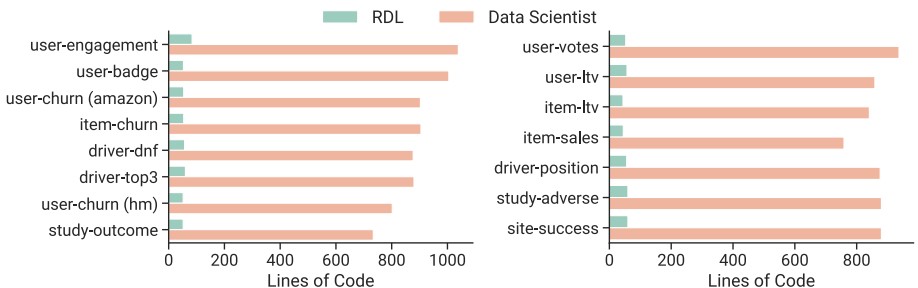

Figure 5: **RDL vs. Data Scientist.** Relational Deep Learning reduces the new lines of code needed to solve a new task by 94%. Left shows entity classification, right shows entity regression.

than an hour to write, whilst the data scientist took 12 hours on average, with a standard deviation of 1.6 hours. We emphasize that this measures *marginal* effort, i.e., it does not include reusable code that can be amortized over many tasks. RDL compares favorably to data scientist because a large majority of RDL code is reusable for new tasks (a GNN architecture and training loop needs only to be defined once) whereas a large portion of the data scientist's code is task specific and must be re-done afresh for every new task that needs to be solved.

**Lines of Code**. Results shown in Figure 5. For the RDL model, the only new addition needed to solve a new task is the code describing how to compute the training supervision for the RDL, which is stored in the training table. This requires a similar number of lines of code for each task, with 56 lines of code on average, with standard deviation $8.8$, with the data scientist requiring with $878 \pm 77$. The minimum lines of code required by RDL is 44, compared to 734 for the data scientist, and maximum is 84 compared to 1039 for the data scientist. Examples of the RDL code for `rel-amazon` tasks can be viewed here. We record the number of lines of data scientist code for EDA and SQL files, and the manipulations needed to format data to be fed into the pre-prepared LightGBM script.

## 6 Related Work

**Graph Machine Learning Benchmarks**. Challenging and realistic benchmarks drive innovation in methodology. A classic example is the ImageNet (Deng *et al.*, 2009), introduced prior to the rise of deep learning, which was a key catalyst for the seminal work of Krizhevsky *et al.* (2017). In graph machine learning, benchmarks such as the Open Graph Benchmark (Hu *et al.*, 2020), TUDataset (Morris *et al.*, 2020), and more recently, the Temporal Graph Benchmark (Huang *et al.*, 2024) have sustained the growth and maturation of graph machine learning as a field. RELBENCH differs since instead of collecting together tasks are already recognized as graph machine learning tasks, RELBENCH presents existing tasks typically solved using other methods, as graph ML tasks. As a consequence, RELBENCH significantly expands the space of problems solvable using graph ML. Whilst graph ML is a key part of this benchmark, relational deep learning is a *new problem*, requiring only need good GNNs, but also innovation on tabular learning to fuse multimodal input data with the GNN, temporal learning, and even graph construction. We believe that advancing the state-of-the-art on RELBENCH will involve progress in all of these directions.

**Relational Deep Learning**. Several works have proposed to use graph neural networks for learning on relational data (Schlichtkrull *et al.*, 2018; Cvitkovic, 2019; Šír, 2021; Zahradník *et al.*, 2023). They explored different graph neural network architectures on (heterogeneous) graphs, leveraging relational structure. Recently, Fey *et al.* (2024) proposed a general end-to-end learnable framework for solving predictive tasks on relational databases, treating temporality as a core concept.

## 7 Conclusion

We introduce RELBENCH, a benchmark for relational deep learning (Fey *et al.*, 2024). RELBENCH provides diverse and realistic relational databases and define practical predictive tasks that cover both entity-level prediction and entity link prediction. In addition, we provide the first open-source implementation of relational deep learning and validated its effectiveness over the common practice of manual feature engineering by an experienced data scientist. We hope RELBENCH will catalyze further research on relational deep learning to achieve highly-accurate prediction over complex multi-tabular datasets without manual feature engineering.

## Acknowledgments and Disclosure of Funding

We thank Shirley Wu, Kaidi Cao, Rok Sosic, Yu He, Qian Huang, Bruno Ribeiro and Michi Yasunaga for discussions and for providing feedback on our manuscript. We also gratefully acknowledge the support of NSF under Nos. OAC-1835598 (CINES), CCF-1918940 (Expeditions), DMS-2327709 (IHBEM); Stanford Data Applications Initiative, Wu Tsai Neurosciences Institute, Stanford Institute for Human-Centered AI, Chan Zuckerberg Initiative, Amazon, Genentech, GSK, Hitachi, SAP, and UCB. The content is solely the responsibility of the authors and does not necessarily represent the official views of the funding entities.

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
