# A  Relational Deep Learning Implementation

As part of RELBENCH, we provide an initial implementation of relational deep learning, based on the blueprint of Fey *et al.* (2024).[4] Our implementation consists four major components: (1) heterogeneous temporal graph, (2) deep learning model, (3) temporal-aware training of the model, and (4) task-specific loss, which we briefly discuss now.

**Heterogeneous temporal graph**. Given a set of tables with primary-foreigh key relations between them we follow Fey *et al.* (2024) to automatically construct a heterogeneous temporal graph, where each table represents a node type, each row in a table represents a node, and a primary-foreign-key relation between two table rows (nodes) represent an edge between the respective nodes. Some node types are associated with time attributes, representing the timestamp at which a node appears. The heterogeneous temporal graph is represented as a PyTorch Geometric graph object. Each node in the heterogeneous graph comes with a rich feature derived from diverse columns of the corresponding table. We use Tensor Frame provided by PyTorch Frame (Hu *et al.*, 2024) to represent rich node features with diverse column types, *e.g.*, numerical, categorical, timestamp, and text.

**Deep learning model**. First, we use deep tabular models that encode raw row-level data into initial node embeddings using PyTorch Frame (Hu *et al.*, 2024) (specifically, we use the ResNet tabular model (Gorishniy *et al.*, 2021)). These initial node embeddings are then fed into a GNN to iteratively update the node embeddings based on their neighbors. For the GNN we use the heterogeneous version of the GraphSAGE model (Hamilton *et al.*, 2017; Fey and Lenssen, 2019) with sum-based neighbor aggregation. Output node embeddings are fed into task-specific prediction heads and are learned end-to-end.

**Temporal-aware subgraph sampling**. We perform temporal neighbor sampling, which samples a subgraph around each entity node at a given seed time. Seed time is the time in history at which the prediction is made. When collecting the information to make a prediction at a given seed time, it is important for the model to only use information from before the seed time and thus not learn from the future (post the seed time). Crucially, when sampling mini-batch subgraphs we make sure that all nodes within the sampled subgraph appear before the seed time (Hamilton *et al.*, 2017; Fey *et al.*, 2024), which systematically avoids time leakage during training. The sampled subgraph is fed as input to the GNN, and trained to predict the target label.

**Task-specific prediction head and loss**. For entity-level classification, we simply apply an MLP on an entity embedding computed by our GNN to make prediction. For the loss function, we use the binary cross entropy loss for entity classification and $L_1$ loss for entity regression.

Recommendation requires computing scores between pairs of source nodes and target nodes. For this task type, we consider two representative predictive architectures: two-tower GNN (Wang *et al.*, 2019) and identity-aware GNN (ID-GNN) (You *et al.*, 2021). First, the two-tower GNN computes the pairwise scores via inner product between source and target node embeddings, and the standard Bayesian Personalized Ranking loss (Rendle *et al.*, 2012) is used to train the two-tower model (Wang *et al.*, 2019). Second, the ID-GNN computes the pairwise scores by applying an MLP prediction head on target entity embeddings computed by GNN for each source entity. The ID-GNN is trained by the standard binary cross entropy loss.

# B  Additional Task Information

For reference, the following list documents all the predictive tasks in RELBENCH.

1. `rel-amazon`
   Node-level tasks:

   (a) `user-churn`: For each user, predict 1 if the customer does not review any product in the next 3 months, and 0 otherwise.

   (b) `user-ltv`: For each user, predict the $ value of the total number of products they buy and review in the next 3 months.

   (c) `item-churn`: For each product, predict 1 if the product does not receive any reviews in the next 3 months.

---

[4]Code available at: https://github.com/snap-stanford/relbench.

(d) `item-ltv`: For each product, predict the $ value of the total number purchases and reviews it recieves in the next 3 months.

Link-level tasks:

(a) `user-item-purchase`: Predict the list of distinct items each customer will purchase in the next 3 months.

(b) `user-item-rate`: Predict the list of distinct items each customer will purchase and give a 5 star review in the next 3 months.

(c) `user-item-review`: Predict the list of distinct items each customer will purchase and give a detailed review in the next 3 months.

2. `rel-avito`

Node-level tasks:

(a) `user-visits`: Predict whether each customer will visit more than one Ad in the next 4 days.

(b) `user-clicks`: Predict whether each customer will click on more than one Ads in the next 4 day.

(c) `ad-ctr`: Assuming the Ad will be clicked in the next 4 days, predict the Click-Through-Rate (CTR) for each Ad.

Link-level tasks:

(a) `user-ad-visit`: Predict the list of ads a user will visit in the next 4 days.

3. `rel-f1`

Node-level tasks:

(a) `driver-position`: Predict the average finishing position of each driver all races in the next 2 months.

(b) `driver-dnf`: For each driver predict the if they will DNF (did not finish) a race in the next 1 month.

(c) `driver-top3`: For each driver predict if they will qualify in the top-3 for a race in the next 1 month.

4. `rel-hm`

Node-level tasks:

(a) `user-churn`: Predict the churn for a customer (no transactions) in the next week.

(b) `item-sales`: Predict the total sales for an article (the sum of prices of the associated transactions) in the next week.

Link-level tasks:

(a) `user-item-purchase`: Predict the list of articles each customer will purchase in the next seven days.

5. `rel-stack`

Node-level tasks:

(a) `user-engagement`: For each user predict if a user will make any votes, posts, or comments in the next 3 months.

(b) `post-votes`: For each user post predict how many votes it will receive in the next 3 months

(c) `user-badge`: For each user predict if each user will receive in a new badge the next 3 months.

Link-level tasks:

(a) `user-post-comment`: Predict a list of existing posts that a user will comment in the next two years.

(b) `post-post-related`: Predict a list of existing posts that users will link a given post to in the next two years.

6. `rel-trial`

Node-level tasks:

(a) `study-outcome`: Predict if the trials in the next 1 year will achieve its primary outcome.

Table 6: Entity classification results (AUROC mean$_{\pm\text{std}}$ over 5 runs, higher is better) on RELBENCH. Best values are in bold along with those not statistically different from it.

| Dataset | Task | Split | LightGBM | RDL |
|---|---|---|---|---|
| rel-amazon | user-churn | Val | $52.05_{\pm0.06}$ | $\mathbf{70.45}_{\pm0.06}$ |
| | | Test | $52.22_{\pm0.06}$ | $\mathbf{70.42}_{\pm0.05}$ |
| | item-churn | Val | $62.39_{\pm0.20}$ | $\mathbf{82.39}_{\pm0.02}$ |
| | | Test | $62.54_{\pm0.18}$ | $\mathbf{82.81}_{\pm0.03}$ |
| rel-avito | user-visits | Val | $53.31_{\pm0.09}$ | $\mathbf{69.65}_{\pm0.04}$ |
| | | Test | $53.05_{\pm0.32}$ | $\mathbf{66.20}_{\pm0.10}$ |
| | user-clicks | Val | $55.63_{\pm0.31}$ | $\mathbf{64.73}_{\pm0.32}$ |
| | | Test | $53.60_{\pm0.59}$ | $\mathbf{65.90}_{\pm1.95}$ |
| rel-event | user-repeat | Val | $\mathbf{67.76}_{\pm0.97}$ | $\mathbf{71.25}_{\pm2.53}$ |
| | | Test | $68.04_{\pm1.82}$ | $\mathbf{76.89}_{\pm1.59}$ |
| | user-ignore | Val | $87.96_{\pm0.28}$ | $\mathbf{91.70}_{\pm0.33}$ |
| | | Test | $79.93_{\pm0.49}$ | $\mathbf{81.62}_{\pm1.11}$ |
| rel-f1 | driver-dnf | Val | $68.42_{\pm1.14}$ | $\mathbf{71.36}_{\pm1.54}$ |
| | | Test | $\mathbf{68.56}_{\pm3.89}$ | $\mathbf{72.62}_{\pm0.27}$ |
| | driver-top3 | Val | $67.76_{\pm2.75}$ | $\mathbf{77.64}_{\pm3.16}$ |
| | | Test | $\mathbf{73.92}_{\pm5.75}$ | $\mathbf{75.54}_{\pm0.63}$ |
| rel-hm | user-churn | Val | $56.05_{\pm0.05}$ | $\mathbf{70.42}_{\pm0.09}$ |
| | | Test | $55.21_{\pm0.12}$ | $\mathbf{69.88}_{\pm0.21}$ |
| rel-stack | user-engagement | Val | $65.12_{\pm0.25}$ | $\mathbf{90.21}_{\pm0.07}$ |
| | | Test | $63.39_{\pm0.26}$ | $\mathbf{90.59}_{\pm0.09}$ |
| | user-badge | Val | $65.39_{\pm0.05}$ | $\mathbf{89.86}_{\pm0.08}$ |
| | | Test | $63.43_{\pm0.12}$ | $\mathbf{88.86}_{\pm0.08}$ |
| rel-trial | study-outcome | Val | $\mathbf{68.30}_{\pm0.53}$ | $\mathbf{68.18}_{\pm0.49}$ |
| | | Test | $\mathbf{70.09}_{\pm1.41}$ | $\mathbf{68.60}_{\pm1.01}$ |

(b) study-adverse: Predict the number of affected patients with severe advsere events/death for the trial in the next 1 year.

(c) site-success: Predict the success rate of a trial site in the next 1 year.

Link-level tasks:

(a) condition-sponsor-run: Predict whether this condition will have which sponsors.

(b) site-sponsor-run: Predict whether this sponsor will have a trial in a facility.

7. rel-event

Node-level tasks:

(a) user-attendance: Predict how many events each user will respond yes or maybe in the next seven days.

(b) user-repeat: Predict whether a user will attend an event(by responding yes or maybe) in the next 7 days if they have already attended an event in the last 14 days.

(c) user-ignore: Predict whether a user will ignore more than 2 event invitations in the next 7 days.

# C Experiment Details and Additional Results

## C.1 Detailed Results

Tables 6, 7 and 8 show mean and standard deviations over 5 runs for the entity classification, entity regression and link prediction results respectively.

## C.2 Hyperparameter Choices

All our RDL experiments were run based on a single set of default task-specific hyperparameters, *i.e.* we did not perform exhaustive hyperparamter tuning, *cf.* Table 9. This verifies the stability and robustness of RDL solutions, even against expert data scientist baselines. Specifically, all task types use a shared GNN configuration (a two-layer GNN with a hidden feature size of 128 and "sum" aggregation) and sample subgraphs identically (disjoint subgraphs of 512 seed entities with a

Table 7: Entity regression results (MAE mean$_{\pm\text{std}}$ over 5 runs, lower is better) on RELBENCH. Best values are in bold along with those not statistically different from it.

| Dataset | Task | Split | Global Zero | Global Mean | Global Median | Entity Mean | Entity Median | LightGBM | RDL |
|---|---|---|---|---|---|---|---|---|---|
| rel-amazon | user-ltv | Val | 14.141 | 20.740 | 14.141 | 17.685 | 15.978 | $14.141_{\pm0.000}$ | $\mathbf{12.132}_{\pm0.007}$ |
| | | Test | 16.783 | 22.121 | 16.783 | 19.055 | 17.423 | $16.783_{\pm0.000}$ | $\mathbf{14.313}_{\pm0.013}$ |
| | item-ltv | Val | 72.096 | 78.110 | 59.471 | 80.466 | 68.922 | $55.741_{\pm0.049}$ | $\mathbf{45.140}_{\pm0.068}$ |
| | | Test | 77.126 | 81.852 | 64.234 | 78.423 | 66.436 | $60.569_{\pm0.047}$ | $\mathbf{50.053}_{\pm0.163}$ |
| rel-avito | ad-ctr | Val | 0.048 | 0.048 | 0.040 | 0.044 | 0.044 | $0.037_{\pm0.000}$ | $\mathbf{0.037}_{\pm0.000}$ |
| | | Test | 0.052 | 0.051 | 0.043 | 0.046 | 0.046 | $\mathbf{0.041}_{\pm0.000}$ | $\mathbf{0.041}_{\pm0.001}$ |
| rel-event | user-attendance | Val | 0.262 | 0.457 | 0.262 | 0.296 | 0.268 | $0.262_{\pm0.000}$ | $\mathbf{0.255}_{\pm0.007}$ |
| | | Test | **0.264** | 0.470 | **0.264** | 0.304 | 0.269 | $\mathbf{0.264}_{\pm0.000}$ | $\mathbf{0.258}_{\pm0.006}$ |
| rel-f1 | driver-position | Val | 11.083 | 4.334 | 4.136 | 7.181 | 7.114 | $3.450_{\pm0.030}$ | $\mathbf{3.193}_{\pm0.024}$ |
| | | Test | 11.926 | 4.513 | 4.399 | 8.501 | 8.519 | $4.170_{\pm0.137}$ | $\mathbf{4.022}_{\pm0.119}$ |
| rel-hm | item-sales | Val | 0.086 | 0.142 | 0.086 | 0.117 | 0.086 | $0.086_{\pm0.000}$ | $\mathbf{0.065}_{\pm0.000}$ |
| | | Test | 0.076 | 0.134 | 0.076 | 0.111 | 0.078 | $0.076_{\pm0.000}$ | $\mathbf{0.056}_{\pm0.000}$ |
| rel-stack | post-votes | Val | 0.062 | 0.146 | 0.062 | 0.102 | 0.064 | $0.062_{\pm0.000}$ | $\mathbf{0.059}_{\pm0.000}$ |
| | | Test | 0.068 | 0.149 | 0.068 | 0.106 | 0.069 | $0.068_{\pm0.000}$ | $\mathbf{0.065}_{\pm0.000}$ |
| rel-trial | study-adverse | Val | 57.083 | 75.008 | 56.786 | 57.083 | 57.083 | $\mathbf{45.774}_{\pm1.191}$ | $46.290_{\pm0.304}$ |
| | | Test | 57.930 | 73.781 | 57.533 | 57.930 | 57.930 | $\mathbf{44.011}_{\pm0.998}$ | $44.473_{\pm0.209}$ |
| | site-success | Val | 0.475 | 0.462 | 0.475 | 0.447 | 0.450 | $0.417_{\pm0.003}$ | $\mathbf{0.401}_{\pm0.009}$ |
| | | Test | 0.462 | 0.468 | 0.462 | 0.448 | 0.441 | $0.425_{\pm0.003}$ | $\mathbf{0.400}_{\pm0.020}$ |

Table 8: Link prediction results (MAP mean$_{\pm\text{std}}$ over 5 runs, higher is better) on RELBENCH. Best values are in bold along with those not statistically different from it.

| Dataset | Task | Split | Global Popularity | Past Visit | LightGBM | RDL (GraphSAGE) | RDL (ID-GNN) |
|---|---|---|---|---|---|---|---|
| rel-amazon | user-item-purchase | Val | 0.31 | 0.07 | $0.18_{\pm0.07}$ | $\mathbf{1.53}_{\pm0.05}$ | $0.13_{\pm0.00}$ |
| | | Test | 0.24 | 0.06 | $0.16_{\pm0.05}$ | $\mathbf{0.74}_{\pm0.08}$ | $0.10_{\pm0.00}$ |
| | user-item-rate | Val | 0.16 | 0.09 | $0.22_{\pm0.02}$ | $\mathbf{1.42}_{\pm0.06}$ | $0.15_{\pm0.00}$ |
| | | Test | 0.15 | 0.07 | $0.17_{\pm0.01}$ | $\mathbf{0.87}_{\pm0.05}$ | $0.12_{\pm0.00}$ |
| | user-item-review | Val | 0.18 | 0.05 | $0.14_{\pm0.03}$ | $\mathbf{1.03}_{\pm0.03}$ | $0.11_{\pm0.00}$ |
| | | Test | 0.11 | 0.04 | $0.09_{\pm0.01}$ | $\mathbf{0.47}_{\pm0.05}$ | $0.09_{\pm0.00}$ |
| rel-avito | user-ad-visit | Val | 0.01 | 3.66 | $0.17_{\pm0.01}$ | $0.09_{\pm0.01}$ | $\mathbf{5.40}_{\pm0.02}$ |
| | | Test | 0.00 | 1.95 | $0.06_{\pm0.01}$ | $0.02_{\pm0.00}$ | $\mathbf{3.66}_{\pm0.02}$ |
| rel-hm | user-item-purchase | Val | 0.36 | 1.07 | $0.44_{\pm0.03}$ | $0.92_{\pm0.04}$ | $\mathbf{2.64}_{\pm0.00}$ |
| | | Test | 0.30 | 0.89 | $0.38_{\pm0.02}$ | $0.80_{\pm0.03}$ | $\mathbf{2.81}_{\pm0.01}$ |
| rel-stack | user-post-comment | Val | 0.03 | 2.05 | $0.04_{\pm0.02}$ | $0.43_{\pm0.08}$ | $\mathbf{15.17}_{\pm0.15}$ |
| | | Test | 0.02 | 1.42 | $0.04_{\pm0.03}$ | $0.11_{\pm0.05}$ | $\mathbf{12.72}_{\pm0.22}$ |
| | post-post-related | Val | 0.47 | 0.00 | $1.62_{\pm0.36}$ | $0.00_{\pm0.01}$ | $\mathbf{7.76}_{\pm0.20}$ |
| | | Test | 1.46 | 1.74 | $2.00_{\pm0.43}$ | $0.07_{\pm0.08}$ | $\mathbf{10.83}_{\pm0.22}$ |
| rel-trial | condition-sponsor-run | Val | 2.63 | 8.58 | $4.88_{\pm0.13}$ | $3.12_{\pm0.24}$ | $\mathbf{11.33}_{\pm0.04}$ |
| | | Test | 2.52 | 8.42 | $4.82_{\pm0.20}$ | $2.89_{\pm0.39}$ | $\mathbf{11.36}_{\pm0.08}$ |
| | site-sponsor-run | Val | 4.91 | 15.90 | $10.92_{\pm0.67}$ | $14.09_{\pm0.77}$ | $\mathbf{17.43}_{\pm0.07}$ |
| | | Test | 3.75 | 17.31 | $8.40_{\pm0.70}$ | $10.70_{\pm1.10}$ | $\mathbf{19.00}_{\pm0.12}$ |

maximum of 128 neighbors for each foreign key). Across task types, we only vary the learning rate and the maximum number of epochs to train for.

Notably, we found that our default set of hyperparameters heavily underperformed on the node-level tasks on the rel-trial dataset. On this dataset, we used a learning rate of 0.0001, a "mean" neighborhood aggregation scheme, 64 sampled neighbors, and trained for a maximum of 20 epochs. For the ID-GNN link-prediction experiments on rel-trial, it was necessary to use a four-layer deep GNN in order to ensure that destination nodes are part of source node-centric subgraphs.

## C.3  Ablations

We also report additional results ablating parts of our relational deep learning implementation. All experiments are designed to be data-centric, aiming to validate basic properties of the chosen datasets and tasks. Examples include confirming that the graph structure, node features, and temporal-awareness all play important roles in achieving optimal performance, which also underscores the unique challenges our RELBENCH dataset and tasks present.

Table 9: Task-specific RDL default hyperparameters.

| Hyperparameter | Task type | | |
| --- | --- | --- | --- |
| | Node classification | Node regression | Link prediction |
| Learning rate | 0.005 | 0.005 | 0.001 |
| Maximum epochs | 10 | 10 | 20 |
| Batch size | 512 | 512 | 512 |
| Hidden feature size | 128 | 128 | 128 |
| Aggregation | summation | summation | summation |
| Number of layers | 2 | 2 | 2 |
| Number of neighbors | 128 | 128 | 128 |
| Temporal sampling strategy | uniform | uniform | uniform |

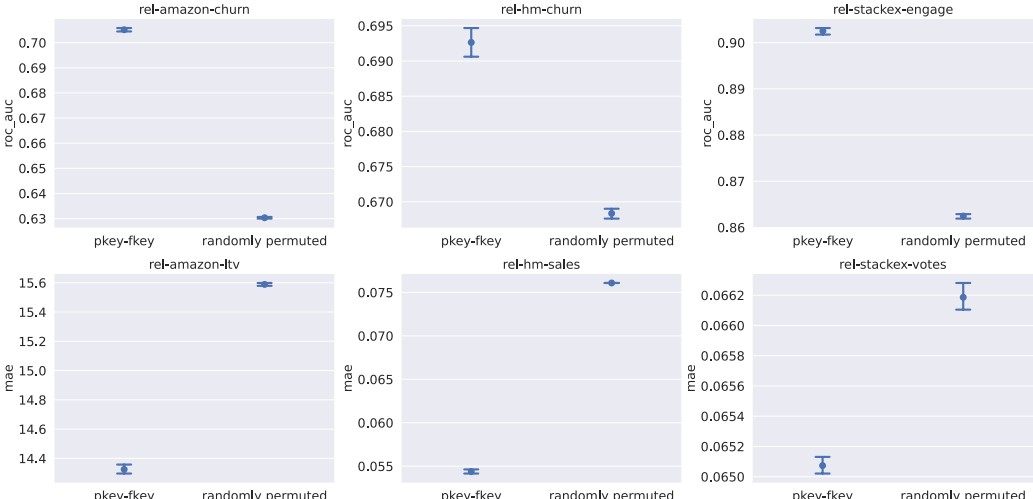

Figure 6: Investigation on the role of leveraging primary-foreign key (pkey-fkey) edges for the GNN. At the top row are three node classification tasks with metric AUROC (higher is better) while at the bottom are three node regression tasks with metric MAE (lower is better), evaluated on the test set. We find that our proposal of using pkey-fkey edges for message passing is vital for GNN to achieve desirable performance on RELBENCH. Error bars correspond to 95% confidence interval.

**Graph structure.** We first investigate the role of the graph structure we adopt for GNNs on REL-BENCH. Specifically, we compare the following two approaches of constructing the edges: **1.** *Primary-foreign key (pkey-fkey)*, where the entities from two tables that share the same primary key and foreign key are connected through an edge; **2.** *Randomly permuted*, where we apply a random permutation on the destination nodes in the primary-foreign key graph for each type of the edge while keeping the source nodes untouched. From Fig. 6 we observe that with random permutation on the primary-foreign key edges the performance of the GNN becomes much worse, verifying the critical role of carefully constructing the graph structure through, *e.g.*, primary-foreign key as proposed in Fey *et al.* (2024).

**Node features and text embeddings.** Here we study the effect of node features used in RELBENCH. In the experiments depicted in Fig. 7, we compare GNN (w/ node feature) with its variant where the node features are all masked by zeros (*i.e.*, w/o node feature). We find that utilizing rich node features incorporated in our RELBENCH dataset is crucial for GNN. Moreover, we also investigate, in particular, the approach to encode texts in the data that constitutes part of the node features. In Fig. 8, we compare GloVe text embedding (Pennington *et al.*, 2014) and BERT text embedding (Devlin *et al.*, 2018) with w/o text embedding, where the text embeddings are masked by zeros. We observe that encoding the rich texts in RELBENCH with GloVe or BERT embedding consistently yields better performance compared with using no text features. We also find that BERT embedding is usually better than GloVe embedding especially for node classification tasks, which suggests that enhancing the quality of text embedding will potentially help achieve better performance.

**Temporal awareness.** We also investigate the importance of injecting temporal awareness into the GNN by ablating on the time embedding. To be specific, in the implementation we add a relative

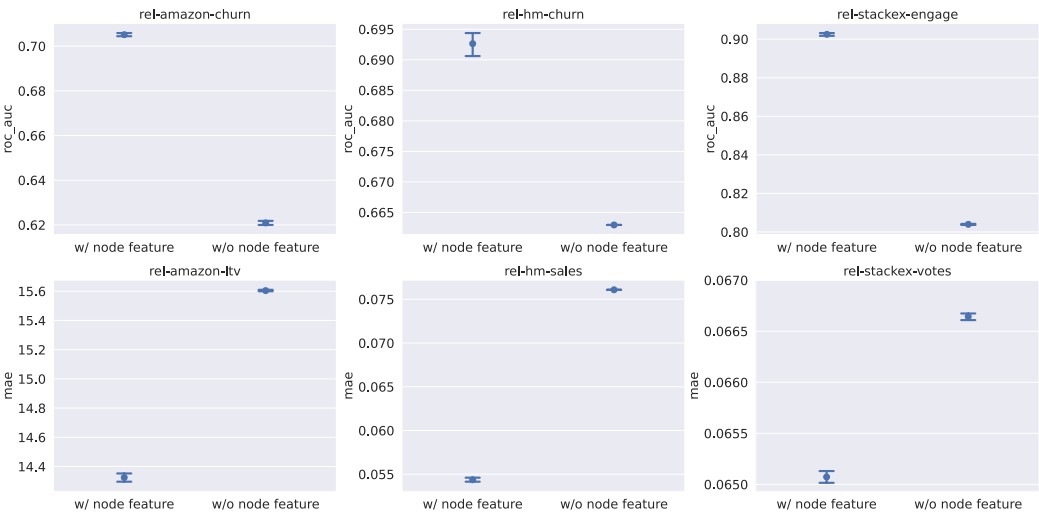

Figure 7: Investigation on the role of node features. At the top row are three node classification tasks with metric AUROC (higher is better) while at the bottom are three node regression tasks with metric MAE (lower is better), evaluated on the test set. We observe that leveraging node features is important for GNN. Error bars correspond to 95% confidence interval.

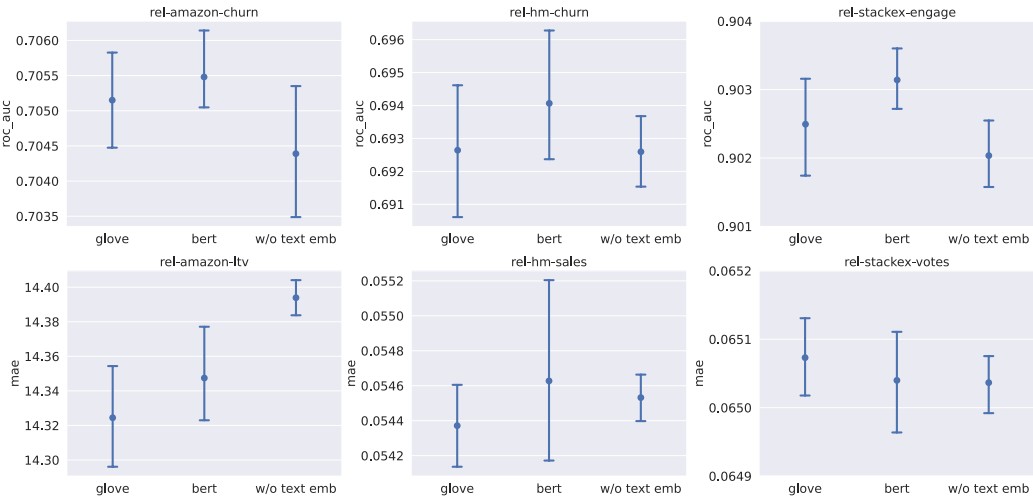

Figure 8: Investigation on the role of text embedding. At the top row are three node classification tasks with metric AUROC (higher is better) while at the bottom are three node regression tasks with metric MAE (lower is better), evaluated on the test set. We observe that adding text embedding using GloVe (Pennington *et al.*, 2014) or BERT (Devlin *et al.*, 2018) generally helps improve the performance. Error bars correspond to 95% confidence interval.

time embedding when deriving the node features using the relative time span between the timestamp of the entity and the querying seed time. Results are exhibited in Fig. 9. We discover that adding the time embedding significantly enhance the performance across a diverse range of tasks, demonstrating the efficacy and importance of building up the temporal awareness into the model.

# D    User Study Additional Details

## D.1    Data Scientist Example Workflow

In this section we provide a detailed description of the data scientist workflow for the `user-churn` task of the `rel-hm` dataset. The purpose of this is to exemplify the efforts undertaken by the

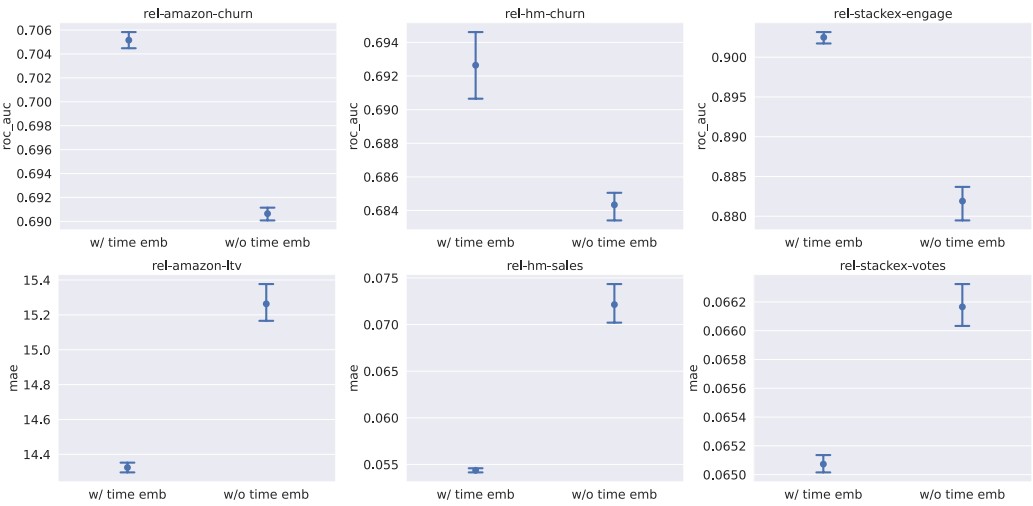

Figure 9: Investigation on the role of time embedding. At the top row are three node classification tasks with metric AUROC (higher is better) while at the bottom are three node regression tasks with metric MAE (lower is better), evaluated on the test set. We find that adding time embedding to the GNN consistently boosts the performance. Error bars correspond to 95% confidence interval.

data scientist to solve RELBENCH tasks. For data scientist solutions to all tasks, see https://github.com/snap-stanford/relbench-user-study.

Recall that the main data science workflow steps are:

1. Exploratory data analysis (EDA).
2. Feature ideation.
3. Feature enginnering.
4. Tabular ML.
5. Post-hoc analysis of feature importance (optional).

### D.1.1 Exploratory Data Analysis

During the exploratory data analysis (EDA) the data scientist familiarizes themselves with a new dataset. It is typically carried out in a Jupyter notebook, where the data scientist first loads the dataset or establishes a connection to it and then systematically explores it. The data scientist may:

- Visualize the database schema, looking at the fields of different tables and the relationships between them.
- Closely analyze the label sets:
  - Look at the relative sizes and temporal split of the training, validation and test subsets.
  - Look at label statistics such as the mean, the standard deviation and various quantiles.
  - For classification tasks, understand class (im)balance: how much bigger is the modal class than the rest? For example, in the user-churn task roughly 82% of the samples have label 1, so there is a good amount of imbalance but not enough to strictly require up-sampling techniques.
  - For regression tasks, understand the label distribution: are the labels concentrated around a typical value or do they follow a power law wherein the labels span several orders of magnitude? In extreme cases, this exploration will point to a need for specialized handing of the label space for model training.
- Plot distributions and aggregations of interesting columns/fields. For example, in Figure 10 we can see three such plots. From left to right:
  - The first plot shows the distribution of age among customers. We see two distinct peaks one in the mid-twenties and another in the mid-fifties, suggesting different customer "archetypes", which may have different spending patterns.

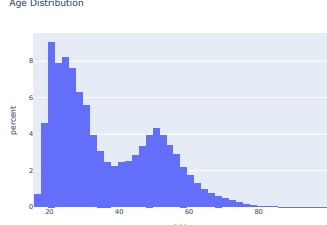
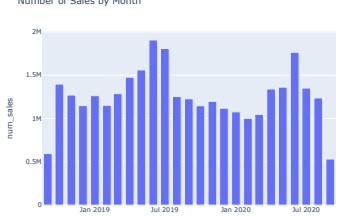
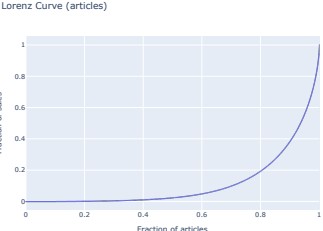

Figure 10: **EDA Plots.** Each plot explores different characteristics of the dataset. Understanding the data and identifying relationships between different quantities is an essential prerequisite to meaningful feature engineering.

   – The second plot shows the number of sales per month over a two year period. We can see some seasonality with summer months being particularly good for overall sales. This suggests date related features could be useful.

   – The third plot shows a *Lorenz curve* of sales per article, showcasing the canonical *Pareto Principle*: 20% of the articles account for 80% of the sales.

• Run custom queries to look at interesting quantities and/or relationships between different columns. For instance, in the EDA for `rel-hm`, an interesting quantity to look at is the variability in item prices across the year. This reveals that most of the variability is downward, representing temporary discounts.

• Investigate outliers or odd-looking patterns in the data. These usually will have some real-world explanation that may inform how the data scientist chooses to pre-process the data and construct features.

In all, this process takes in the order of a few hours (3-4 for most datasets in the user study).

### D.1.2 Feature Ideation

Having explored the dataset in the EDA, the data scientist will then brainstorm features that, to their judgement, will provide valuable signal to a model for a specific learning task. In the case of the `user-churn` task, a rather simple feature would be the customer's age, which is a field directly available in one of the tables. A slightly more complex feature would be the total amount spent by the customer so far. Finally, an example of a fairly complex feature is the average monthly sales volume of items purchased by the customer in the past week. A high value for this feature may indicate that the customer has been shopping trendy items lately, whereas a low value for this feature may indicate that the customer has been interested in more arcane or specific items.

In practice, the ideation phase consists of writing down all of these feature ideas in a file or a piece of paper. It is the quickest part of the whole process and in this user study took between 30 minutes and one hour.

### D.1.3 Feature Engineering

With a list of features in hand, the data scientist then proceeds to actually write code to generate all the features for each sample in the the train, validation and test subsets. In this user study, this was carried out using DuckDB SQL[5] with some Jinja templating[6] for convenience.

Revisiting the example features from the previous section, the conceptual complexity of the features closely tracks with the technical complexity of implementing them. For customer age all that is required is a simple *join*. The total amount spent by the customer, can be calculated using a *group by* clause and a couple of *join*'s. Lastly, calculating the average monthly sales volume of items purchased by the customer in the past week requires multiple *group by*'s, *join*'s, and *window functions* distributed across multiple *common table expressions* (CTEs).

---

[5]See https://duckdb.org/.
[6]See https://jinja.palletsprojects.com/en/3.1.x/intro/.

A key consideration during feature engineering is the prevention of *leakage*. The data scientist must ensure that none of the features accidentally include information from after the sample timestamp. This is especially true for complex features like the third example above, where special care must be taken to ensure that each *join* has the appropriate filters to comply with the sample timestamp.

For some tasks, *e.g.*, `study-outcome`, the initial features *did* leak information from the validation set into the training set. Thanks to the RELBENCH testing setup, leaking test data into the training data is hard to do by accident, since test data is hidden. Leaking information from validation to train (but not test to train) led to extremely high validation performance and very low test performance (test was significantly lower than LightGBM with no feature engineering). The large discrepancy between validation and test performances alerted the data scientist to the mistake, and the features were eventually fixed. This example illustrates another complexity that feature engineering introduces, with special care needed to ensure leakage does not happen.

Other considerations that the data scientist must keep in mind during development and implementation of the features are parsing issues, runtime constrains and memory load. For example, during the user study we identified a parsing issue arising from special characters in user posts/comments in the `rel-stack` dataset. The *backslash* character, widely used LaTeXcan trip up certain text parsers if not handled with care. Furthermore, runtime and memory constraints are important to keep in mind when working with larger datasets and computing features that require nested *join*'s and aggregations. During the user study, there were some cases where we had to refactor SQL queries to make them more efficient, increasing the overall implementation time. For some tasks we had to implement sub-sampling of the training set to reduce the burden on compute resources.

Finally, once the features have been generated for each data subset, the data scientist will usually inspect the generated features looking for anomalies (e.g. an unusual prevalence of *NULL* values). In this user study we also implemented some automated sanity checks to validate the generated features beyond manual inspection.

### D.1.4  Tabular Machine Learning

The output of the Feature Engineering phase is a DuckDB table with engineered features for each data subset. There is some non-trivial amount of work required to go from those tables to the numerical arrays used for training by most Tabular ML models (LightGBM in this case). This is implemented in a Python script that loads the data, transforms it into arrays and carries out hyperparameter tuning. In this user study we ran 5 hyperparameter optimization runs, with 10 trials each, reporting the mean and standard deviation over the 5 runs. For the `user-churn` task this took one to two hours.

### D.1.5  Post-hoc Analysis

The last step in the process is to look at a trained model and analyze its performance and feature importance. To this end we used SHAP values (Lundberg and Lee, 2017) and the corresponding python package[7]. Figure 11 shows the top 30 most important features in the `user-churn` task. The individual *violin plots* show the distribution of SHAP values for a subset of the validation set, the color indicates the value of the feature. For the `user-churn` task, the most predictive features were primarily (1) all-time statistics of user behavior pattern, and (2) temporal information that allows the model to be aware of seasonality.

### D.2  Regression Output Head Analysis

By default, our RDL implementation uses a simple linear output head on top of the GNN embeddings. However we found that on regression tasks this sometimes led to lower than desirable performance. We found that performance on many regression tasks could be improved by modifying this output head. Instead of a linear layer, we took the output from the GNN, and fed these embeddings into a LightGBM model, which is trained in a second separate training phase from the GNN model.

The resulting model still uses an end-to-end learned GNN for cross-table feature engineering, showing that the GNN is learning useful features. Instead we attribute the weaker performance to the linear output head. We believe that further attention to the regression output head is an interesting direction for further study, with the goal of designing an output head that is performant and can be trained jointly with the GNN (unlike our LightGMB modification).

---

[7]See https://shap.readthedocs.io/en/latest/.

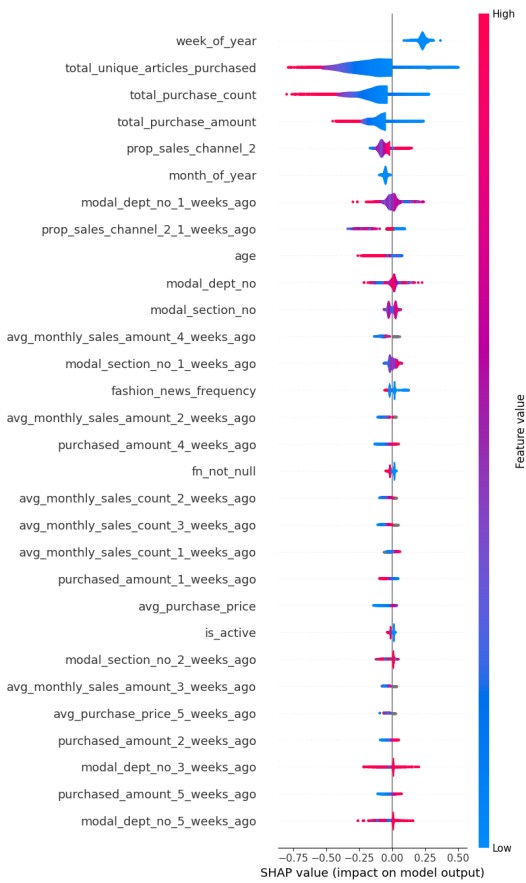

Figure 11: **Feature Importances.** SHAP values of top 30 features ranked by importance. Note: *week_of_year* feature shows little variability because the validation set is temporally concentrated in a few weeks.

We run three experiments to study this phenomena, and attempt to isolate the output head as a problematic component for regression tasks.

1. LightGBM trained on GNN-learned entity-level features on regression tasks. We find that this model performs better than the original GNN, suggesting that the linear output head of the GNN is suboptimal.

2. LightGBM trained on GNN-learned entity-level features on classification tasks. We find no performance improvement, and even some degradation, compared to the original GNN model, suggesting that the observed performance boost of (1) comes not from an overall better architecture but from the correction of an innate shortcoming of the linear output head vis-a-vis regression tasks. In other words, using a LightGBM on top of the GNN is only helpful insofar as it provides a more flexible output head for regression tasks.

3. Evaluate GNN performance after converting regression tasks to binary classification tasks with label $y = \mathbf{1}\{y_{\text{regression}} > 0\}$. We find that the performance gap between the data scientist models and the GNN narrow. This suggests that the GNN can learn the relevant predictive signal, but performance is affected by how the task is formulated (classification vs regression).

See Tables 10, 11, 12 for the results of each of these experiments.

In Figure 3, for regression tasks we report the RDL results using GNN learned features with LightGBM output head. In Table 4 we report result for the basic GNN in order to avoid creating confusion for other researchers when comparing different GNN methods. We believe that Tables

Table 10: Entity regression results (MAE, lower is better) on selected RELBENCH datasets. Training a LightGBM model on features extracted by a trained GNN leads to performance lift. This is evidence that the linear layer output head of the base GNN is suboptimal.

| Dataset | Task | Split | GNN | GNN+LightGBM |
|---|---|---|---|---|
| rel-f1 | driver-position | Test | $4.173_{\pm 0.178}$ | $\mathbf{4.05}_{\pm 0.09}$ |
| rel-stack | post-votes | Test | $0.065_{\pm 0.00}$ | $\mathbf{0.062}_{\pm 0.00}$ |
| rel-amazon | item-ltv | Test | $14.31_{\pm 0.028}$ | $\mathbf{14.10}_{\pm 0.02}$ |

Table 11: Entity classification results (AUROC, higher is better, numbers bolded if withing standard deviation of best result) on selected RELBENCH tasks. Training a LightGBM model on features extracted by a trained GNN does not lead to performance lift, and can even hurt performance slightly. This is evidence that output head limitations hold for regression tasks only. Note, study-outcome uses default GNN parameters for simplicity, differing form the performance reported in the main paper.

| Dataset | Task | Split | GNN | GNN+LightGBM |
|---|---|---|---|---|
| rel-f1 | driver-dnf | Test | $\mathbf{72.3}_{\pm 1.67}$ | $\mathbf{71.8}_{\pm 1.30}$ |
| rel-trial | study-outcome | Test | $\mathbf{68.8}_{\pm 1.10}$ | $\mathbf{68.2}_{\pm 0.44}$ |
| rel-stack | user-badge | Test | $88.3_{\pm 0.04}$ | $\mathbf{88.4}_{\pm 0.04}$ |

10, 11, 12 provide clear evidence that there is an opportunity for improvements and simplifications, which we leave to future work.

Table 12: Entity classification results (AUROC, higher is better) on selected RELBENCH regression tasks, converted into classification tasks with binary label $y = \mathbf{1}\{y_{\text{regression}} > 0\}$. Training a LightGBM model on features extracted by a trained GNN leads to performance lift. This is evidence that the linear layer output head of the base GNN is suboptimal.

| Dataset | Task | Split | GNN | Data Scientist |
|---|---|---|---|---|
| rel-f1 | driver-position | Test | $81.96_{\pm 1.18}$ | $\mathbf{86.63}_{\pm 0.40}$ |
| rel-stack | post-votes | Test | $\mathbf{80.5}_{\pm 0.18}$ | $78.3_{\pm 0.05}$ |
| rel-amazon | item-ltv | Test | $\mathbf{70.61}_{\pm 0.06}$ | $70.29_{\pm 0.06}$ |

# E   Dataset Origins and Licenes

This section details the sources for all data used in RELBENCH. In all cases, the data providers consent for their data to be used freely for non-commercial and research purposes. The only database with potentially personally identifiable information is rel-stack, which draws from the Stack Exchange site, which sometimes has individuals' names as their username. This information shared with consent, as all users must agree to the Stack Exchange privacy policy, see: https://stackoverflow.com/legal/privacy-policy.

**rel-amazon**. Data obtained from the Amazon Review Data Dump from Ni *et al.* (2019). See the website: https://cseweb.ucsd.edu/~jmcauley/datasets/amazon_v2/. Data license is not specified.

**rel-avito**. Data is obtained from Kaggle https://www.kaggle.com/competitions/avito-context-ad-clicks. All RELBENCH users must download data from Kaggle themselves, a part of which is accepting the data usage terms. These terms include use only for non-commercial and academic purposes. Note that after data download, we further downsample the avito dataset by randomly selecting approximately 100,000 data point from user table and sample all other tables that have connections to the sampled users.

**rel-stack**. Data was obtained from The Internet Archive, whose stated mission is to provide "universal access to all knowledge. We downloaded our data from https://archive.org/download/stackexchange in Novermber 2023. Data license is not specified.

**rel-f1**. Data was sourced from the Ergast API (https://ergast.com/mrd/) in February 2024. The Ergast Developer API is an experimental web service which provides a historical record of

motor racing data for non-commercial purposes. As far as we are able to determine the data is public and license is not specified.

**rel-trial**. Data was downloaded from the `ClinicalTrials.gov` website in January 2024. This data is provided by the NIH, an official branch of the US Government. The terms of use state that data are available to all requesters, both within and outside the United States, at no charge. Our `rel-trial` database is a snapshot from January 2024, and will not be updated with newer trials results.

**rel-hm**. Data is obtained from Kaggle `https://www.kaggle.com/competitions/h-and-m-personalized-fashion-recommendations`. All RELBENCH users must download data from Kaggle themselves, a part of which is accepting the data usage terms. These terms include use only for non-commercial and academic purposes.

**rel-event**. The dataset employed in this research was initially released on Kaggle for the Event Recommendation Engine Challenge, which can be accessed at `https://www.kaggle.com/c/event-recommendation-engine-challenge/data`. We have obtained explicit consent from the creators of this dataset to use it within RELBENCH. We extend our sincere gratitude to Allan Carroll for his support and generosity in sharing the data with the academic community.

# F    Additional Training Table Statistics

We report additional training table statistics for all tasks, separated into entity classification (*cf.* Table 13), entity regression (*cf.* Table 14), and link prediction (*cf.* Table 15).

Table 13: RELBENCH entity classification training table target statistics.

| Dataset | Task | Split | Positives | Negatives |
|---|---|---|---|---|
| rel-amazon | user-churn | Train | 2,956,658 (62.47%) | 1,775,897 (37.53%) |
| | | Val | 263,098 (64.2%) | 146,694 (35.8%) |
| | | Test | 213,400 (60.64%) | 138,485 (39.36%) |
| | item-churn | Train | 1,113,863 (43.52%) | 1,445,401 (56.48%) |
| | | Val | 73,242 (41.22%) | 104,447 (58.78%) |
| | | Test | 61,647 (36.95%) | 105,195 (63.05%) |
| rel-f1 | driver-dnf | Train | 1,365 (11.96%) | 10,046 (88.04%) |
| | | Val | 125 (22.08%) | 441 (77.92%) |
| | | Test | 207 (29.49%) | 495 (70.51%) |
| | driver-top3 | Train | 231 (17.07%) | 1,122 (82.93%) |
| | | Val | 119 (20.24%) | 469 (79.76%) |
| | | Test | 128 (17.63%) | 598 (82.37%) |
| rel-hm | user-churn | Train | 3,170,367 (81.89%) | 701,043 (18.11%) |
| | | Val | 62,225 (81.28%) | 14,331 (18.72%) |
| | | Test | 61,609 (82.61%) | 12,966 (17.39%) |
| rel-stack | user-engagement | Train | 68,020 (5.0%) | 1,292,830 (95.0%) |
| | | Val | 2,411 (2.81%) | 83,427 (97.19%) |
| | | Test | 2,411 (2.74%) | 85,726 (97.26%) |
| | user-badge | Train | 163,048 (4.81%) | 3,223,228 (95.19%) |
| | | Val | 7,301 (2.95%) | 240,097 (97.05%) |
| | | Test | 6,735 (2.64%) | 248,625 (97.36%) |
| rel-trial | study-outcome | Train | 7,647 (63.76%) | 4,347 (36.24%) |
| | | Val | 561 (58.44%) | 399 (41.56%) |
| | | Test | 483 (58.55%) | 342 (41.45%) |
| rel-event | user-repeat | Train | 1,882 (48.98%) | 1,960 (51.02%) |
| | | Val | 130 (48.51%) | 138 (51.49%) |
| | | Test | 110 (44.72%) | 136 (55.28%) |
| | user-ignore | Train | 3,247 (16.88%) | 15,992 (83.12%) |
| | | Val | 441 (10.54%) | 3,744 (89.46%) |
| | | Test | 450 (11.40%) | 3,499 (88.60%) |
| rel-avito | user-clicks | Train | 2,302 (3.87%) | 57,152 (96.13%) |
| | | Val | 745 (3.52%) | 20,438 (96.48%) |
| | | Test | 740 (1.54%) | 47,256 (98.46%) |
| | user-visits | Train | 78,467 (90.59%) | 8,152 (9.41%) |
| | | Val | 27,086 (90.35%) | 2,893 (9.65%) |
| | | Test | 30,731 (85.06%) | 5,398 (14.94%) |

Table 14: RELBENCH entity regression training table target statistics.

| Dataset | Task | Split | Minimum | Median | Mean | Maximum |
|---------|------|-------|---------|--------|------|---------|
| rel-amazon | user-ltv | Train | 0.0 | 0.0 | 16.93 | 9,511.46 |
| | | Val | 0.0 | 0.0 | 14.14 | 7,259.91 |
| | | Test | 0.0 | 0.0 | 16.78 | 10,329.86 |
| | | Total | 0.0 | 0.0 | 16.71 | 10,329.86 |
| | item-ltv | Train | 0.0 | 20.78 | 67.57 | 198,419.8 |
| | | Val | 0.0 | 22.44 | 72.10 | 75,901.55 |
| | | Test | 0.0 | 23.72 | 77.13 | 206,663.58 |
| | | Total | 0.0 | 20.97 | 68.38 | 206,663.58 |
| rel-f1 | driver-position | Train | 1.0 | 13.33 | 13.90 | 39.0 |
| | | Val | 1.0 | 11.4 | 11.08 | 22.0 |
| | | Test | 1.0 | 12.18 | 11.93 | 24.0 |
| | | Total | 1.0 | 13.0 | 13.57 | 39.0 |
| rel-hm | item-sales | Train | 0.0 | 0.0 | 0.076 | 87.16 |
| | | Val | 0.0 | 0.0 | 0.086 | 40.36 |
| | | Test | 0.0 | 0.0 | 0.076 | 38.31 |
| | | Total | 0.0 | 0.0 | 0.076 | 87.16 |
| rel-stack | post-votes | Train | 0.0 | 0.0 | 0.093 | 78.0 |
| | | Val | 0.0 | 0.0 | 0.062 | 36.0 |
| | | Test | 0.0 | 0.0 | 0.068 | 26.0 |
| | | Total | 0.0 | 0.0 | 0.090 | 78.0 |
| rel-trial | study-adverse | Train | 0.0 | 2.0 | 39.84 | 28,085.0 |
| | | Val | 0.0 | 2.0 | 57.08 | 17,245.0 |
| | | Test | 0.0 | 3.0 | 57.93 | 5,978.0 |
| | | Total | 0.0 | 2.0 | 42.20 | 28,085.0 |
| | site-success | Train | 0.0 | 0.0 | 0.44 | 1.0 |
| | | Val | 0.0 | 0.4 | 0.47 | 1.0 |
| | | Test | 0.0 | 0.17 | 0.4 | 1.06 |
| | | Total | 0.0 | 0.0 | 0.45 | 1.0 |
| rel-event | user-attendance | Train | 0.0 | 0.0 | 0.37 | 16.0 |
| | | Val | 0.0 | 0.0 | 0.28 | 5.0 |
| | | Test | 0.0 | 0.0 | 0.26 | 8.0 |
| | | Total | 0.0 | 0.0 | 0.34 | 16.0 |
| rel-avito | ad-ctr | Train | 0.00052 | 0.018 | 0.045 | 1.0 |
| | | Val | 0.00091 | 0.018 | 0.048 | 1.0 |
| | | Test | 0.00085 | 0.019 | 0.052 | 1.0 |
| | | Total | 0.00052 | 0.018 | 0.047 | 1.0 |

Table 15: RELBENCH link prediction training table link statistics.

| Dataset | Task | Split | #Links | Avg #links per entity/timestamp | %Repeated links |
|---|---|---|---|---|---|
| rel-amazon | user-item-purchase | Train | 11,759,844 | 2.18 | - |
| | | Val | 802,540 | 2.28 | 0.18 |
| | | Test | 918,919 | 2.33 | 0.15 |
| | user-item-rate | Train | 7,146,115 | 1.8 | - |
| | | Val | 519,496 | 2.01 | 0.19 |
| | | Test | 599,867 | 2.05 | 0.15 |
| | user-item-review | Train | 5,138,184 | 2.19 | - |
| | | Val | 268,651 | 2.3 | 0.18 |
| | | Test | 305,476 | 2.4 | 0.15 |
| rel-hm | user-item-purchase | Train | 13,191,321 | 3.38 | - |
| | | Val | 237,152 | 3.18 | 3.51 |
| | | Test | 207,996 | 3.10 | 3.76 |
| rel-stack | user-post-comment | Train | 43,337 | 2.08 | - |
| | | Val | 1,603 | 1.94 | 3.43 |
| | | Test | 1,517 | 2.0 | 4.09 |
| | post-post-related | Train | 7,162 | 1.2 | - |
| | | Val | 294 | 1.3 | 0.0 |
| | | Test | 359 | 1.39 | 1.39 |
| rel-trial | condition-sponsor-run | Train | 503,176 | 12.51 | - |
| | | Val | 30,448 | 14.63 | 34.48 |
| | | Test | 25,694 | 12.49 | 38.37 |
| | site-sponsor-run | Train | 1,485,360 | 2.27 | - |
| | | Val | 80,103 | 2.16 | 20.91 |
| | | Test | 50,635 | 1.85 | 23.29 |
| rel-avito | user-ad-visit | Train | 2,738,733 | 31.53 | - |
| | | Val | 877,441 | 29.27 | 6.79 |
| | | Test | 712,985 | 19.73 | 4.73 |

# G  Dataset Schema

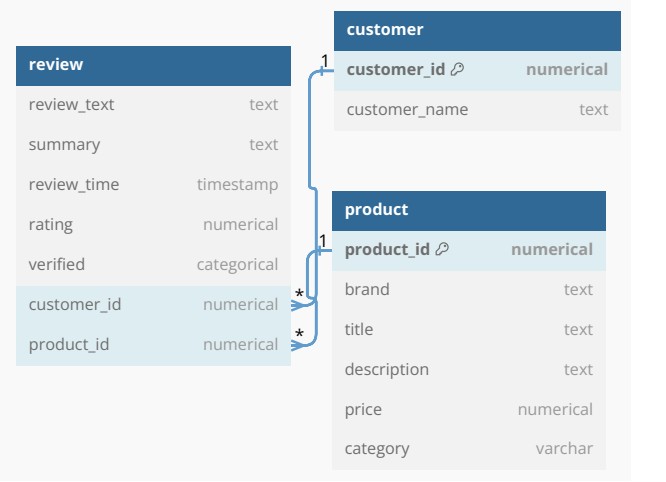

Figure 12: rel-amazon database diagram.

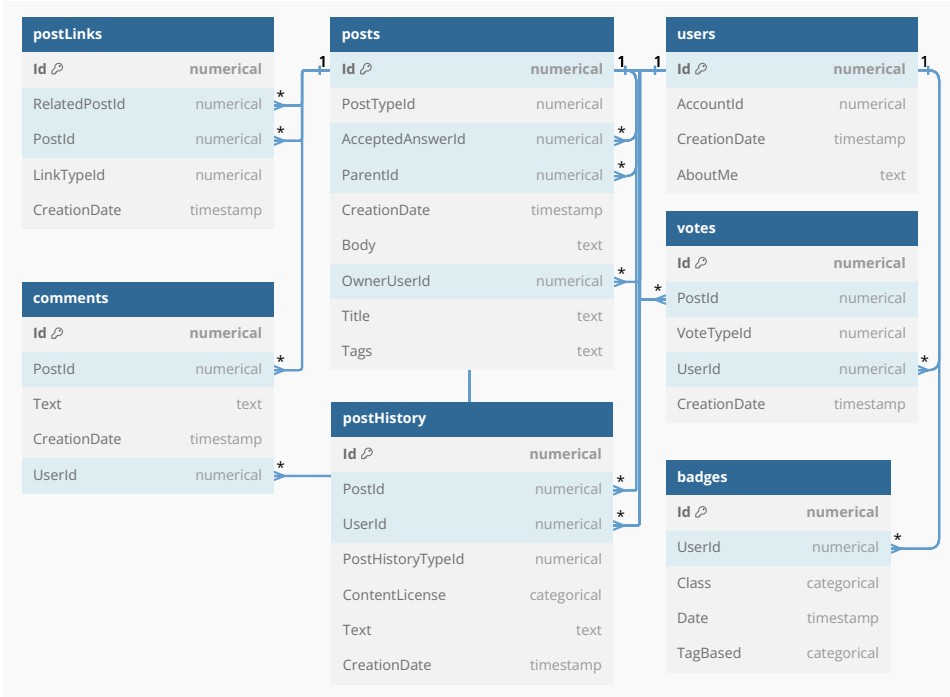

Figure 13: `rel-stack` database diagram.

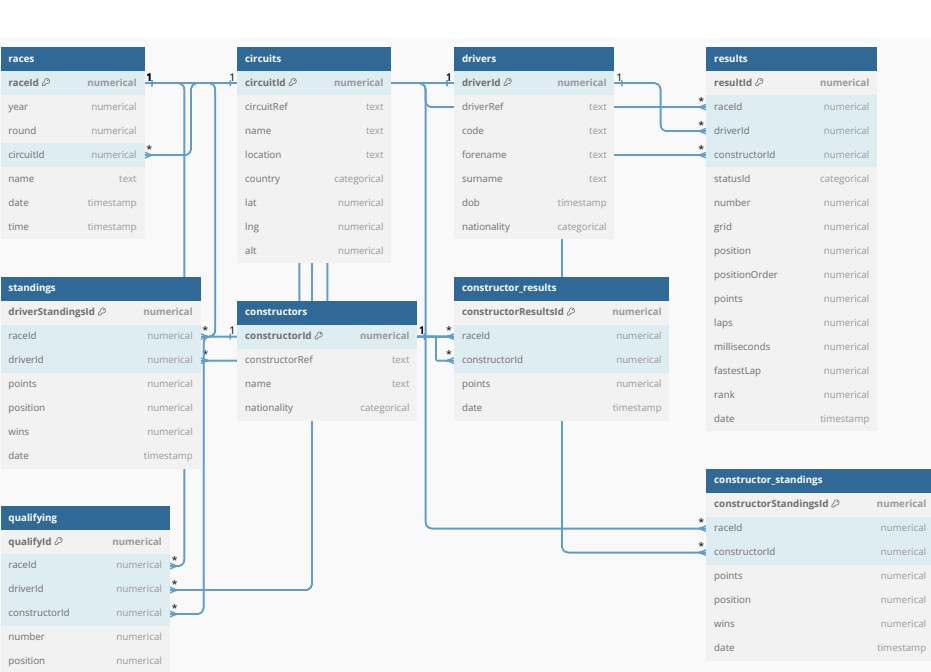

Figure 14: `rel-f1` database diagram.

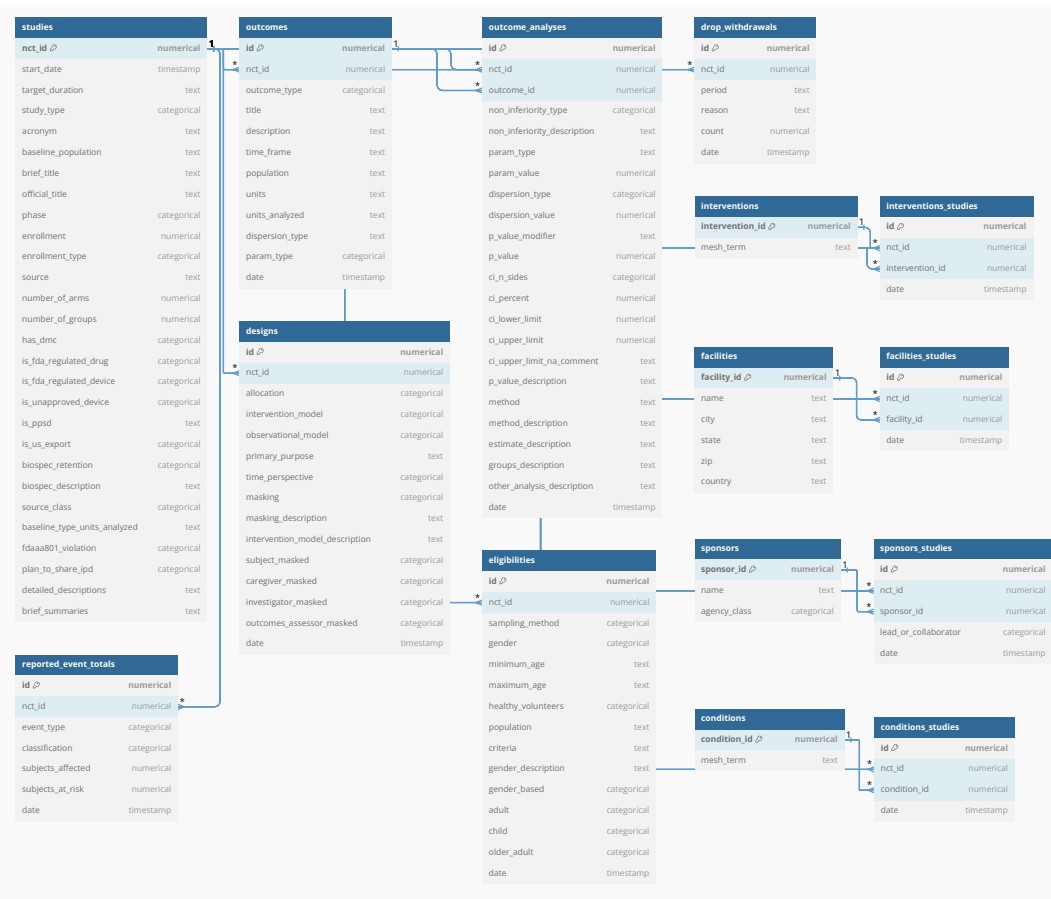

Figure 15: `rel-trial` database diagram.

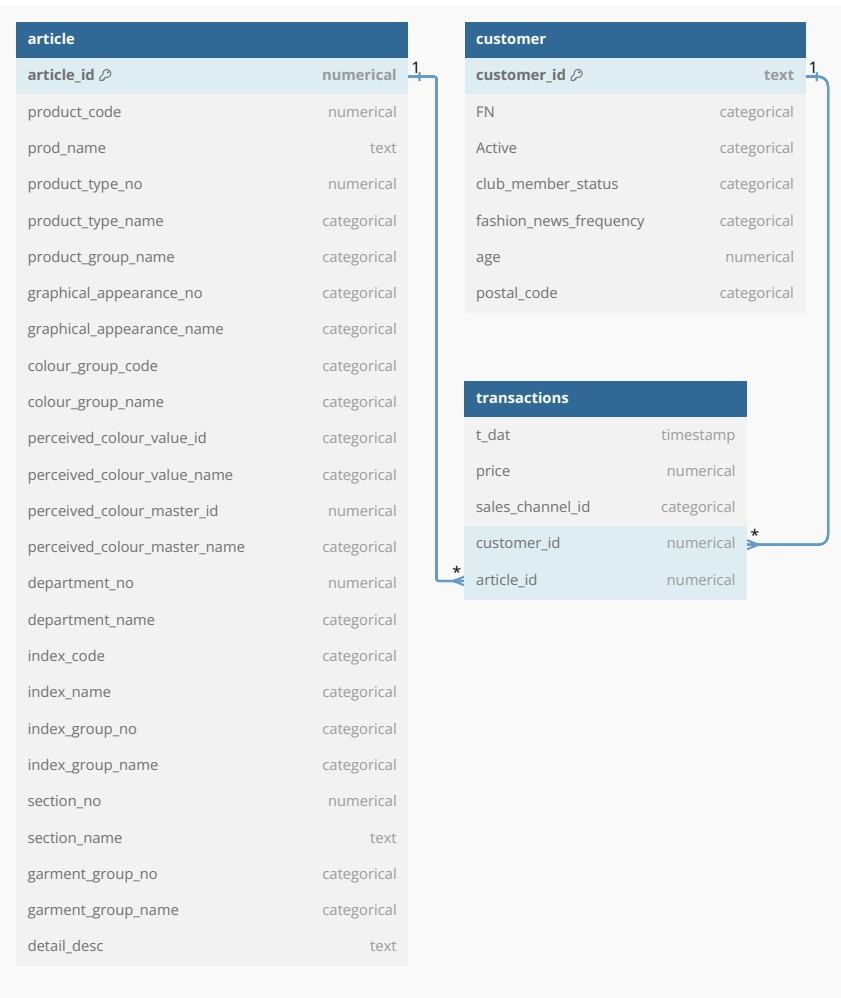

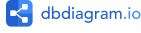

Figure 16: `rel-hm` database diagram.

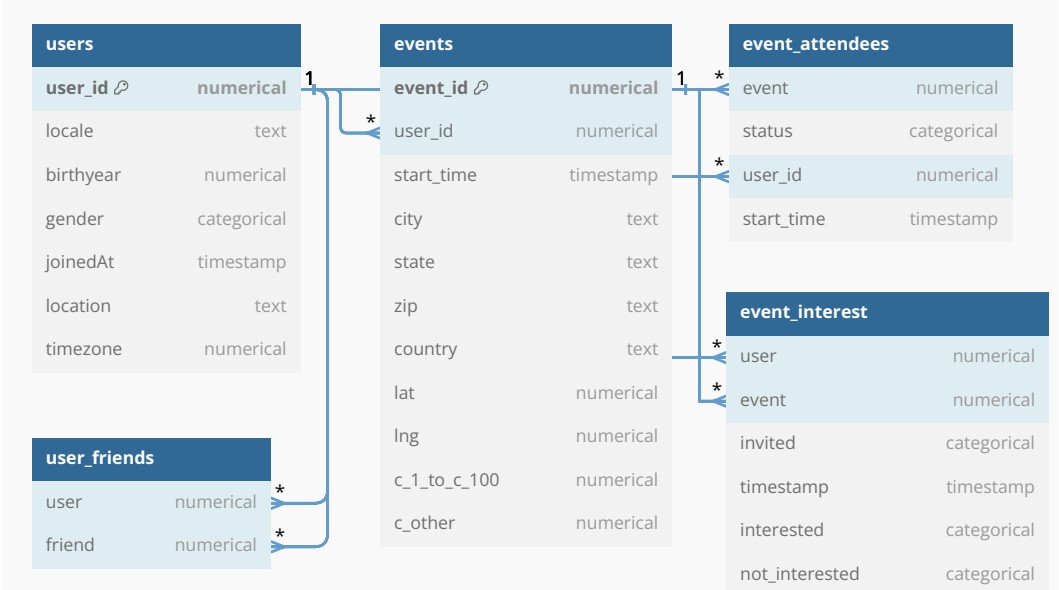

Figure 17: `rel-event` database diagram.

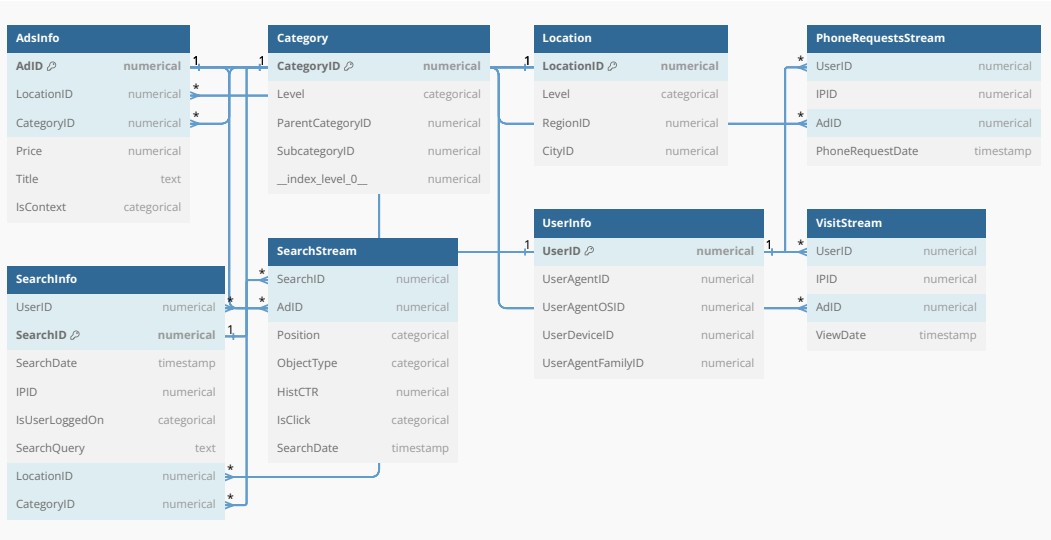

Figure 18: `rel-avito` database diagram.

## H  Additional Discussion

As well as datasets and model benchmarking, the RELBENCH package aims to make the use of Relational Deep Learning on other problems not considered in this work. Here we discuss several key usability choices make during the design of the RELBENCH package.

**Adding new datasets is easy**. RELBENCH organizes datasets as subclasses of a parent dataset class. An example definition for `rel-stack` can be found here. This class reads the raw data (e.g., from csv or parquet) into pandas data frame format and stores a dictionary of all the database tables. It also stores a small amount of metadata to indicate which column is the primary key column, which (if any) are foreign-keys linking to another table, and which (if any) column is a timestamp column. All other class functionality is standardized in the parent dataset class, allowing maximal compatibility with the model training and minimal effort to define a new dataset class.

Once defined and added to the database registry, a user can load the data using RELBENCH utility function

```
dataset = get_dataset("dataset-name")
```

and run RELBENCH model training scripts on the data with no further modifications required. Importantly RELBENCH is designed so that this functionality is outward facing—i.e., a user can define their own new dataset and add it to the registry without modifying the package source code. We wrote a tutorial of how to do this here. In practice, we found that writing a new dataset class took no more than a couple of hours.

**Data Processing**. Models are trained using databases essentially as they are found as this is one of the key promises of Relational Deep Learning. That said, one important pre-processing that was necessary is to drop columns which introduce temporal leakage. This happens when a cell in a row is updated *after* the row is first created, adding information from after the associated timestamp. For example, the raw Stack Exchange database includes a "was answered" TRUE/FALSE column for each question. We deleted this column since providing this as input to any model (RDL or otherwise) would leak information that wouldn't be available at test time. Overwriting of cells is common in real-world relational databases which negatively affects both RDL and data scientist modeling. Designing methods to detect or address this type of temporal leakage would be an interesting direction for future work.

**Hardware usage**. RELBENCH datasets were chosen to be academic-lab friendly, meaning that all training runs can be run on small GPUs e.g., a 11GB NVIDIA GeForce RTA 2080 Ti released in 2018. We also experimented with Quadro RTX 8000 (48GB) and A100s (80GB). In all cases, loading data to launch an experiment takes no more than 1 minute even for the largest RELBENCH dataset. The one-time text embedding cost is the only significant pre-processing expense. This takes at most 40 minutes for the slowest database, which in this case is the `rel-amazon` database because it has a lot of text data including product descriptions and reviews. We found the computation requirements extremely manageable on modest hardware, and provided the data scientist access to the same hardware for their work. For model training, we found that RDL model training never took more than about 2 hours. In many cases the RDL model trained faster than the data scientist model, primarily because the data scientist model included a comprehensive LightGBM hyperparmeter sweep, whereas the RDL models used a single set of RDL hyperparamers with no tuning.

**Reliability of RDL Models**. One advantage RDL models forego in comparison to data scientist designed models is the natural interpretability, and associated trust, that comes from features specified through SQL queries. SQL queries often have natural semantic meaning the give insights into possible failure modes of the model.

An intriguing possibility for future work is to view RDL models as compositions of SQL queries, thereby matching the interpretability standards of data scientist-built models. Intuitively the GNN message passing closely resembles a SQL JOIN + AGGREGATE operation. Indeed, since our graph construction connects a node $v$ (i.e., DB entity) to all entities $v'$ with $v_{\text{pkey}} = v'_{\text{fkey}}$, the GNN message passing propagates information from all such $v'$ to $v$. This is the same as a SQL inner join operation, which makes a new table with a row for all $(v, v')$ pairs, which is often followed by an aggregation (e.g., sum or mode) over $v'$ to get a table with one row for each $v$. This connection points to a possibility to reverse engineer a set of SQL operations matching each GNN layer.

# I   Broader Impact

Relational deep learning broadens the applicability of graph machine learning to include relational databases. Whilst the blueprint is general, and can be applied to a wide variety of tasks, including potentially hazardous ones, we have taken steps to focus attention of potential positive use cases. Specifically, the beta version of RELBENCH considers two databases, Amazon products, and Stack Exchange, that are designed to highlight the usefulness of RDL for driving online commerce and online social networks. Future releases of RELBENCH will continue to expand the range of databases into domains we reasonably expect to be positive, such as biomedical data and sports fixtures. We hope these concrete steps ensure the adoption of RDL for purposes broadly beneficial to society.

Whilst we strongly believe the RELBENCH has all the ingredients needed to be a long term benchamrk for relational deep learning, there are also possibilities for improvement and extension. Two such possibilities include: (1) RDL at scale: currently our implementation must load the entire database into working memory during training. For very large datasets this is not viable. Instead, a custom batch sampler is needed that acesses the database via queries to sample specific entities and their pkey-fkey neighbors; (2) Fully inductive link-prediction: our current link-prediction implementation supports predicting links for test time pairs (*head,tail*) where *head* is potentially new (unseen during training) and *tail* seen in the training data. Extending this formulation to be fully inductive (*i.e.*, *tail* unseen during training) is possible, but out of the scope of this work for now.