# OpenReview forum: "RelBench: A Benchmark for Deep Learning on Relational Databases"
_NeurIPS.cc/2024/Datasets_and_Benchmarks_Track — NeurIPS 2024 Track Datasets and Benchmarks Poster_

### Official Review · Reviewer_jCCf · 2024-06-27
**Review submission 1351**

**Rating:** 9
**Confidence:** 4

**Review:**

This is a good submission with some important flaws.

RelBench (as a collection of RDBs with associated prediction tasks) is on its own a large contribution to any kind of research on relational data (not necessarily related to GNNs or RDL), thanks in large part to the effort that was made to prepare it for reproducibility. The variety in sizes, tasks, data types will be useful to test how methods are affected by each dimension in various ways.

The experimental section clearly shows that RDL is very effective at addressing those prediction tasks, and that it is a promising research direction.

However, various factors undermine the value of this submission.
- The baselines used in the experimental study are not very strong, and prior work is not tested or discussed in detail.
- There is very little detail on the actual implementation of the RDL infrastructure and the resources required to run RDL.
- The lack of details on how the RDL infrastructure is implemented, and on how it is used in the user study undermine that section of the experimental section. As a result, the claim that "RDLs reduce the amount of work required by an order of magnitude" is unconvincing.

**Strengths:**

- Having access to a collection of relational databases with associated predictive tasks is a major contribution that has many applications beyond testing RDLs.
- The experimental section is exhaustive and highlights the effectiveness of relational deep learning on the given tasks. A lot of detail on the experiments is provided in the additional material.
- Experiments also show that RDLs can match the performance of a human analyst, while (given proper preparation) being much faster. Providing information on the case study is also a plus.

**Additional Feedback:**

While I am aware that the current review is quite negative, I do think that RelBench (as a dataset of RDBs with associated predictive tasks) is a very good contribution on its own; I also think that RDLs show clear promise and the experiments clearly highlight their effectiveness.

However, I do not believe that this submission in its current form presents the results fairly and convincingly. Indeed, some of the claims are not supported by evidence, and critical information is missing.

This said, I am more than willing to raise my evaluation if the authors address my main concerns (o5, o3, o2) satisfactorily.

**Clarity:**

The paper is mostly clear and well written, but it is wordy at times and there some typos.

**Correctness:**

Some of the claims are well substantiated by the experiments. The user study is not presented in a convincing way, and the lack of information on resource requirements makes it hard to form a complete evaluation of all methods under consideration.

**Documentation:**

The additional material contains information about the datasets, and further information is available on the website provided in the submission. Additional information on the preprocessing required to prepare the available databases would be useful.

A license is available on the repository.

I was able to install the package. I was not able to find how to re-run experiments, or a clear description of how the RDL framework is implemented. This would make reproducing the results very difficult.

**Ethics:**

No ethical concerns.

**Limitations:**

A further review of prior work that details the reasons for (not) using certain baselines would help with reinforcing the effectiveness of RDL on the predictive tasks that are under consideration. Ideally, at least a simple baseline should be provided and compared against to strengthen the case for using RDL instead.

As already mentioned in previous sections, the user study with the current framing is unconvincing and does not provide evidence for the argument that is being made.

More detail on the computational resources should be provided, as it would help with understanding the tradeoffs involved in using RDLs and it would help researchers with understanding the physical requirements for using the dataset.

**Opportunities For Improvement:**

o1) The presentation could be improved. In some sections it is too wordy or detailed, and there are typos and poorly formulated sentences. A lot of the detail in table 3 (like the exact number of rows for each training step) could be moved to the appendix; similarly, most of the detail about the DBs could be moved to the appendix too.

o2) Baselines are not very convincing, and prior work is not sufficient. There is no mention of graph embedding methods that are not based on GNNs such as [1] or [2], or other methods that consider the problem of converting relational data to graph format ([3], [4]). I understand that running completely new baselines at this stage may be impractical; still, alternative solutions should be discussed in the prior work section, at least to explain why they would not be applicable to the problems under evaluation, or why they were not considered in the first place. Comparing the preparation of a simple baseline (and its potential downsides) with what is required to run the RDL framework would make the contribution more convincing and the case for using RDL rather than simpler methods more compelling.

o3) There is very little detail on the actual implementation of the RDL infrastructure, even accounting for the additional material. It is not clear if the RDL must be built ad-hoc for any given RDB, or if some components may be re-used. The resources required to run RDL (GPUs, RAM, compute time, CPUs, etc.) are not provided anywhere, aside from mentioning that "the entire database must be loaded in memory to run RDL". Please add more detail on this, and ideally add an estimation of the resources required by the data scientist. This would allow to have a better view of the potential tradeoffs between the effectiveness of RDLs and their cost.

o4) Please add to the appendix some detail on the pre-processing (if any) required to go from the "raw" version of the RDBs to the version that is being used in RelBench.

o5) I find the argument that "Relational Deep Learning learns better models whilst reducing human work by an order of magnitude" not convincing given the evidence provided in the experimental section. From my understanding, the comparison is made by contrasting the time spent by a data scientist working on each benchmark RDB from scratch, with the time spent by a user that re-uses the already prepared RelBench infrastructure to run experiments on the prepared RDBs.
If this is the case, then the message is not "RDL reduces work by an order of magnitude (in general)", but "on RelBench databases, RDL reduces work by an order of magnitude".
If I have misunderstood the argument, then I strongly recommend to be more explicit and rephrase the scenario so that the correct message is conveyed.
How much time would it take to preprocess a RDB that is unknown to both the data scientist and a RDL developer so that the same suite of experiments can be conducted? What's the human effort involved in building the RDL infrastructure?
The experiments and text do not answer these questions. At the moment, what transpires from the experiments is that the suite of code that is provided with the submission allows to save a large amount of work and time _on the RDBs provided_, but it does not provide a compelling argument that this would be the case for _any RDB_.
Either, provide better evidence that the performance of RDLs can be extended to unknown RDBs by describing in better detail the effort involved in preparing the RDL infrastructure (possibly by running a new experiment to show the difference in effort starting from scratch), or revise the argument to clarify that the time save applies only to the RDBs in RelBench.

[1] |Grover, Aditya, and Jure Leskovec. "node2vec: Scalable feature learning for networks." _Proceedings of the 22nd ACM SIGKDD international conference on Knowledge discovery and data mining_. 2016.|
|APA||

[2] |Chen, Haochen, et al. "Harp: Hierarchical representation learning for networks." _Proceedings of the AAAI conference on artificial intelligence_. Vol. 32. No. 1. 2018.|
|APA||

[3]  Cappuzzo, Riccardo, Paolo Papotti, and Saravanan Thirumuruganathan. "Creating embeddings of heterogeneous relational datasets for data integration tasks." _Proceedings of the 2020 ACM SIGMOD international conference on management of data_. 2020.

[4] Cvetkov-Iliev, Alexis, Alexandre Allauzen, and Gaël Varoquaux. "Relational data embeddings for feature enrichment with background information." _Machine Learning_ 112.2 (2023): 687-720.

**Relation To Prior Work:**

Prior work is not very exhaustive. In particular, no attention was given to graph embedding methods that are not based on GNNs, such as [1] or [2], and to RDL alternatives for converting relational data to graph form such as [3] or [4]. The revised version should at least include an explanation of why such baselines could not be applied to the problem at hand.

[1] |Grover, Aditya, and Jure Leskovec. "node2vec: Scalable feature learning for networks." _Proceedings of the 22nd ACM SIGKDD international conference on Knowledge discovery and data mining_. 2016.|
|APA||

[2] |Chen, Haochen, et al. "Harp: Hierarchical representation learning for networks." _Proceedings of the AAAI conference on artificial intelligence_. Vol. 32. No. 1. 2018.|
|APA||

[3]  Cappuzzo, Riccardo, Paolo Papotti, and Saravanan Thirumuruganathan. "Creating embeddings of heterogeneous relational datasets for data integration tasks." _Proceedings of the 2020 ACM SIGMOD international conference on management of data_. 2020.

[4] Cvetkov-Iliev, Alexis, Alexandre Allauzen, and Gaël Varoquaux. "Relational data embeddings for feature enrichment with background information." _Machine Learning_ 112.2 (2023): 687-720.

**Summary And Contributions:**

The submission presents RelBench, a dataset collecting relational databases with associated predictive tasks that allows to benchmark RDB solutions such as relational deep learning. The benchmarking RDBs are then used to evaluate the performance of relational deep learning against various baselines and present a case study comparing RDL against an expert data scientist.

---

> ### Author Rebuttal · Authors · 2024-08-17
>
> Thank you for your review. We are glad that you recognize
> that introducing a collection of relational databases and predictive tasks is a “major contribution”. You raised a number of concerns which we are keen to address.
>
> ---
>
> > I find the argument that RDL reduces human work by an order of magnitude unconvincing. It seems the comparison is made between the data scientist working with the data from scratch vs RDL assuming the data is already loaded.
>
> This is an important point of clarification. **We do not include the data loading/processing time for either RDL or the data scientist.** We only claim that RDL speeds up model development, not data preparation/loading. To establish this we only measure the feature engineering (for data scientist) + model training time to solve a new task on a pre-processed database.
>
> In fact, the data loading pipeline is shared between RDL and the data scientist, so RDL does not introduce any significant overheads for data loading/preparation over a data scientist's approach. We believe that accelerating model development (apart from data loading) is valuable in many use cases where engineers need to solve many different predictive tasks over a single database.
>
> Thank you for raising this. We will clarify our main claim in the manuscript and include a wider discussion of the conclusions that can be drawn from our work. We believe this addresses your key concern over our user study claims, but if further discussion or amendments are needed then please do say. We are very keen to resolve this particular point in a mutually agreeable way.
>
> ---
>
> > There is not enough detail on the RDL data loading infrastructure. It is not clear if the RDL must be built ad-hoc for each new database.
>
> We are happy to clarify this in the revised manuscript.
>
> RelBench provides a unified class structure
> to express datasets and tasks.
> Once defined and added to the registry,
> a dataset can be loaded
> using the utility function `get_dataset("<dataset-name>")`.
> The RDL training scripts can then be run
> without further modifications.
> **This functionality is outward facing --
> users can define their own new datasets/tasks
> without modifying the RelBench source code.** We provide a tutorial [here
> ](https://colab.research.google.com/github/snap-stanford/relbench/blob/main/tutorials/custom_dataset.ipynb). Typically, writing a new dataset class takes no more than a couple of hours.
>
> RelBench caches the one-off data processing that happens
> the first time a dataset is used, allowing for fast data loading.
> This includes graph construction and feature embedding.
> In practice, the main one-off overhead comes from the use of text embedding models
> for text data such as Stack Exchange posts.
>
> ---
>
> > Please add some detail on any preprocessing applied to raw data.
>
> We do minimal preprocessing, as a key selling point of RDL is that it can process databases essentially as they are found.
>
> That said, one important preprocessing that was necessary was to drop columns which introduce temporal leakage.
> This happens when a cell in a row is updated *after*
> the row is first created, adding information from after the associated timestamp.
> For example, the raw Stack Exchange database includes a “was answered” TRUE/FALSE column for each question. We deleted this column since providing it input to a model (RDL or otherwise) would leak information that wouldn’t be available at test time.
>
> Overwriting of cells is common in real-world relational databases
> which negatively affects both RDL and data scientist modeling.
> Designing methods to detect or address
> this type of temporal leakage
> would be an interesting direction for future work.
>
> We will add this discussion to the paper.
>
> ---
>
> > No discussion of baseline choices. In particular no discussion of why we don’t compare to embedding methods such as node2vec.
>
> Many common node embedding methods are not appropriate because RelBench data is temporal. Methods like node2vec, DeepWalk, etc. produce a single embedding for each node, which would leak future information when training on past instances.
>
> Instead we view the key baseline to be the data scientist, since this reflects the prior gold standard ML approach for relational data. This is why we prioritized strengthening this baseline as much as possible.
>
> We believe that this discussion is highly valuable and important to have in the paper. We will update future versions of the paper to include this. Thank you for raising this point and helping us improve our paper.
>
> ---
>
> > The resources required to run RDL (GPUs, RAM, compute time, CPUs, etc.) are not provided anywhere
>
> We designed RelBench datasets to be academic-lab friendly, meaning that all training runs can be run on small GPUs e.g., a 11GB NVIDIA GeForce RTA 2080 Ti released in 2018. We also experimented with Quadro RTX 8000 (48GB) and A100s (80GB). In all cases and datasets, loading data to launch an experiment takes at most 1 minute.
>
> The one-time text embedding cost is the only significant pre-processing expense, taking at most 40 minutes for the slowest database, which in this case is the Amazon database because it has a lot of text data.
>
> **We found the computation requirements extremely manageable on modest hardware, and provided the data scientist access to the same hardware for their work.**
>
> Training an RDL model never took more than about 2 hours. In many cases the RDL model trained faster than the data scientist model, primarily because the data scientist model included a comprehensive LightGBM hyperparmeter sweep, whereas the RDL models used a single set of RDL hyperparamers with no tuning. The RDL hyperparameters were kept the same across all tasks and datasets (except two tasks where we lightly tuned them) to demonstrate the robustness of the RDL model.
>
> We will add this additional discussion to the paper appendix.
>
> ---
> > The presentation could be improved.
>
> Thank you for this feedback. We will thoroughly proofread and edit the manuscript.

---

> > ### Comment · Reviewer_jCCf · 2024-08-20
> >
> > I thank the authors for addressing my comments.
> >
> > I find the response to point 1) convincing. Given the additional context, the comparison between the two settings appears fair and lends credibility to the argument. While I was not doubting the effectiveness of RDL in principle ("in the given scenario, RDL is better"), the original framing appeared to put the human expert in a far worse position by comparison, which undermined the validity of the experimental section.
> >
> > The response to point 2) also helps with addressing this concern, and shows that a substantial effort was put into improving reproducibility by providing exhaustive documentation and the possibility to extend the benchmark with user-provided DBs.
> >
> > 3) This (and point 2) address my comment.
> >
> > 4) The argument is fair. I believe that having the discussion of why such baselines cannot be applied to the problem at hand would reinforce that a more complex architecture is required. Ignoring them gives instead the _impression_ that they have been deliberately ignored to push competing strategies, which is why I raised the issue.
> >
> > I am particularly satisfied with point 5), and with the effort put into building a dataset that can be run in (relatively) low-resource environments. In fact, I even consider the (relatively) small footprint of RelBench as a strong point of the contribution, as it will definitely improve the potential reach of this work. Indeed, one of my main concerns with the RDL architecture was specifically the potential cost of running GNNs.
> >
> > Overall, I am satisfied with the responses from the authors, and I will be positively updating my evaluation as a result. This is a really good contribution.

---

> > > ### Author Response · Authors · 2024-08-21
> > >
> > > We are very glad that this discussion has been useful, and are sincerely grateful for your active engagement in this process.
> > >
> > > The points raised in your review are important and of interest to all readers, so we will update our manuscript accordingly to include all this information. Thank you again for helping us to improve our paper and ensuring that we provide readers with a comprehensive discussion.

---

### Official Review · Reviewer_EgX3 · 2024-07-16
**Benchmark for relational data tasks**

**Rating:** 6
**Confidence:** 4
**Correctness:** Yes, mostly correct from my point of …
**Clarity:** The paper is well written to me.

**Review:**

The motivation for the paper to provide a benchmark for relational data tasks seems rather reasonable.
Overall the paper is well written and the benchmarks and tasks described well.
One main concern is the amount of data sets and the diversity - 6 relational tables is not a lot and that one can do times based splits for all does not really support diversity (not all data is aligned by time).
Existing work such as [1] has already 3 other relational benchmarks (from Kaggle: KDD Cup 2014, Outbrain, Grupo Bimbo). There is some processing needed, but it remains a question to me if the diversity and amount of datasets is sufficient for a proper benchmark.

The user study illustrates the impact of automatically leveraging relational data nicely. One open question there is - how do the automatically generated transforms look like? Are they as understandable as the sql in feat.sql created by Data Scientists?

[1] "One button machine for automating feature engineering in relational databases", https://arxiv.org/pdf/1706.00327

**Strengths:**

Selected data sets and tasks are well described.
A user study is always quite some effort and it is good to get some signal from actual users.

**Additional Feedback:**

None.

**Documentation:**

To me the data is very well organized on the relevant webpage.
It comes with examples and overview of the schema.

**Ethics:**

No.

**Limitations:**

Yes.

**Opportunities For Improvement:**

More data sets, more diversity. In addition, there is too much focus on time - otherwise it is more a relational benchmark for multi-variate timeseries. Time is actually also making it easier for Deep Learning/transformer models, because there is a 'sequence to align' to.

**Relation To Prior Work:**

I think it would not hurt to cite some existing work like [1] above and [2] below which are clearly relevant.

[2] Neural Feature Learning From Relational Database, https://arxiv.org/abs/1801.05372

**Summary And Contributions:**

The paper introduces 6 benchmarks and various tasks (e.g. entity classification) on relational data. The main motivation is to provide a benchmark for evaluating deep learning approaches. The paper explains the data sets in detail as well as the tasks. It compares a baseline approach RDL and a standard performance on the main table using GBT. The work also includes a user study to show case how automatically leveraging relational data information can improve performance even when compared to humans. The paper comes with a clean presentation of the data/tasks on a webpage.

---

> ### Author Rebuttal · Authors · 2024-08-17
>
> Thank you for your time taken to review our work. We are sincerely grateful for your efforts, and are glad that your overall assessment is positive. Your review raised a number of points that we would like to address.
>
> ---
>
> > The user study illustrates the impact of automatically learning features with GNNs. One question is what do these learned features look like?
>
> This is a really excellent question. Although outside the scope of our current effort, we have good reasons to believe that this direction would be very fruitful. Your intuition that the RDL model features could be viewed as SQL queries is exactly correct.
>
> The reason for this is that the GNN message passing closely resembles a SQL JOIN + AGGREGATE operation. Specifically, since our graph construction connects a node $v$ (i.e., DB entity) to all entities $v’$ with $v.pkey = v’.fkey$, the GNN message passing propagates information from all such $v’$ to $v$. This matches a SQL inner join operation, which makes a new table with $(v,v’)$ rows, which is often followed by an aggregation (e.g., sum or mode) over $v’$ to get a table with one row for each $v$.
>
> We believe that the connection between SQL and GNN message passing could be made mathematically formal, pointing to a possibility to reverse engineer a set of SQL operations matching each GNN layer. We believe that interpreting GNNs as SQL operations would be a highly novel interpretability method with exciting potential for big impact.
>
> We will add a discussion of this to the paper, aiming to highlight this research opportunity to the reader.
>
> ---
>
> > One concern is the amount of datasets and the diversity
>
> We respectfully disagree with this assessment. We went through significant efforts requiring a large team to curate 7 diverse databases (updated from 6 at time of submission), covering e-commerce, healthcare, social networks, and sport. Each database has multiple associated tasks, totalling 30 tasks across all databases.
>
> Beyond the datasets themselves, each task was carefully designed to be challenging but for which predictive signal was available. For example, for node-level tasks, we carefully studied the ground truth train/val label distributions to ensure a reasonable skew, and for link-level tasks we adjusted task definitions until the density of links was high enough, in order to ensure plenty of predictive signal for users. This process took many months of dedicated effort from 5+ team members.
>
> We sincerely hope you can appreciate the significance of the effort required to undertake this project.
>
> ---
>
> > RelBench focuses only on time-based databases and data splits, is this diverse enough?
>
> We would like to highlight that most relational databases include some timestamp columns, and hence are temporal as in RelBench. Indeed, the three examples you shared from Kaggle have timestamp columns in some tables, and hence should be treated as temporal. Furthermore, 2 of 3 cases the train/val/test splits are done temporally as in RelBench (the third, Outbrains, did not provide clear documentation on how splits were generated).
>
> These three databases are excellent examples of interesting relational databases, and we believe they demonstrate well the generality of our temporal formulation. All three databases/tasks could be naturally added to RelBench, following the tutorial we wrote  [here](https://colab.research.google.com/github/snap-stanford/relbench/blob/main/tutorials/custom_dataset.ipynb), and solved using RDL.
>
> Finally, although RelBench does not provide any non-temporal databases, 1) users can easily add their own non-temporal databases by following our tutorial, and 2) RelBench model training can be easily modified to support no temporality by simply commenting out one line ([see here](https://github.com/snap-stanford/relbench/blob/main/examples/gnn_node.py#L111)) in the dataloader.
>
> ---
>
> > Related work from Lam et al.
>
> Thank you for sharing this related work! We will add it to the paper along with a comparative discussion.

---

### Official Review · Reviewer_1DEG · 2024-07-26
**Bordenline Accept**

**Rating:** 6
**Confidence:** 4
**Correctness:** NA
**Clarity:** NA

**Review:**

NA

**Strengths:**

1. RELBENCH provides a wide range of databases and predictive tasks from diverse domains such as e-commerce, medical, and social media, which ensures broad applicability and robust evaluation of deep learning models.

2. The paper offers an open-source implementation of RDL, making it accessible for researchers and practitioners to build upon and contribute to the advancement of this field.

3. Through an extensive empirical study, the paper validates the effectiveness of RDL, providing strong evidence that end-to-end deep learning can fully leverage the predictive power of relational databases.

**Additional Feedback:**

NA

**Documentation:**

NA

**Opportunities For Improvement:**

1. While the paper provides an open-source implementation, the complexity of converting relational databases to graph structures and training GNNs might still pose a barrier to entry for some researchers and practitioners.

2.Although RELBENCH includes diverse domains, the ability of the RDL model to generalize to entirely new and unseen domains remains unclear. The paper could include experiments on cross-domain generalization to address this limitation .

3.I wonder how can the system further integrate human expertise to improve model interpretability and trust, particularly in critical applications such as healthcare or finance. There should be a detailed discussion.

**Relation To Prior Work:**

NA

**Summary And Contributions:**

The paper presents RELBENCH, a benchmark for applying deep learning to predictive tasks over relational databases using Graph Neural Networks (GNNs). RELBENCH includes diverse databases, predictive tasks, and an open-source implementation for Relational Deep Learning (RDL). This benchmark aims to facilitate future research and development in the field of relational deep learning.

---

> ### Author Rebuttal · Authors · 2024-08-17
>
> We are grateful for your time taken to review our work, and are very glad that your overall assessment is positive. Your review raised a number of points that we would like to address.
>
> ---
>
> > Will the complexity of converting relational databases into graphs and training GNNs be a barrier to users?
>
> Thanks, this is a great question! We agree that this process is complex and a barrier to entry for researchers. One of the key contributions of RelBench is to provide code designed to minimize this barrier as much as possible, enabling researchers to use RelBench to explore this direction.
>
> The three steps necessary to train models on new datasets/tasks are (1) define the new dataset/task, (2) convert to a graph, and (3) train GNN. For part (1), we designed RelBench so this functionality is outward facing - i.e., **a RelBench user can define their own new dataset/task without modifying the RelBench source code.** We wrote a tutorial of how to do this [here](https://colab.research.google.com/github/snap-stanford/relbench/blob/main/tutorials/custom_dataset.ipynb).  In practice, we found that writing a new dataset class could be done within a few hours. Parts (2) and (3) have fully generic implementations in RelBench, which can be directly reused to train models on any new datasets and tasks.
>
> In all, RelBench is both a collection of standardized datasets and tasks, and a model training library that can be used on new datasets and tasks. We will update the paper to emphasize the dual purpose of RelBench.
>
> ---
>
> > I wonder how the interpretability and trust of the system can be improved, for instance using human expertise.
>
> This is a great excellent question. We have good reasons to believe that interpretability will be a very fruitful direction. We believe that RDL models could be viewed as compositions of SQL queries, thereby matching the interpretability standards of data scientist-built models.
>
> The reason for this is that the GNN message passing closely resembles a SQL JOIN + AGGREGATE operation. Specifically, since our graph construction connects a node $v$ (i.e., DB entity) to all entities $v’$ with $v.pkey = v’.fkey$, the GNN message passing propagates information from all such $v’$ to $v$. This is the same as a SQL inner join operation, which makes a new table with a row for all $(v,v’)$ pairs, which is often followed by an aggregation (e.g., sum or mode) over $v’$ to get a table with one row for each $v$.
>
> This connection points to a possibility to reverse engineer a set of SQL operations matching each GNN layer. We believe that interpreting GNNs as SQL operations would be a highly novel interpretability method with the potential for big impact.
>
> We will add a discussion of this to the paper, aiming to highlight this research opportunity to the reader.
>
> ---
>
> > Can RDL models generalize to unseen domains?
>
> For RelBench our aim was simply to establish the efficacy of RDL in the standard supervised learning one-task-one-model setting. That said, we agree that strong cross-domain generalization is an exciting and important direction, and a very natural next step which we are actively exploring.
>
> Cross-domain generalization raises a number of interesting research questions. For domain generalization, a single trained model must be able to process entirely different databases with different tables and different columns. Equivalently, we need a GNN able to generalize to (1) new relation types, and (2) new input feature spaces. Standard GNNs do not support either of these, pointing to a need for new GNN methodological innovations and new research opportunities.
>
> We look forward to subsequent work developing methods that facilitate study of cross-domain generalization.

---

### Official Review · Reviewer_Vo6o · 2024-07-28
**A benchmark for evaluatating Relational Deep Learning methods**

**Rating:** 8
**Confidence:** 4
**Correctness:** Yes
**Clarity:** Yes

**Review:**

This paper presents a public benchmark for evaluating Relational Deep Learning (RDL) methods. RDL is a new problem, proposed by some of the same authors of this paper earlier this year, that focuses on solving predictive tasks over relational databases with graph neural networks. This benchmark has the potential to accelerate research on RDL: It consists of a broad, representative set of databases spanning multiple domains (e-commerce, Q&A platforms, medicine and sports) and a wide variety of tasks. The authors also publicly release a pipeline for evaluation of the methods and keeping a leaderboard; evaluate their previously proposed method for RDL using this benchmark; and compare against the results obtained by an experienced data scientist doing manual feature engineering (gold standard prior to RDL). The evaluation shows that RDL outperforms the data scientist’s models in most of the cases whilst reducing human effort by 96%.

A limitation of the work is that the baselines used for entity classification and entity regression (other than the comparison against the data scientist) seem weak as they are not geared towards the RDL problem. Furthermore, the user study is limited to a single data scientist and thus the results are anecdotal in nature. While the data scientist is very experience in general, it is not clear if they have any particular expertise in the domains/tasks at hand. It would be interesting to understand what data scientists with deep expertise in those domains would produce and how that would compare against RDL.

**Strengths:**

- The benchmark has the potential to accelerate research on RDL: It consists of a broad, representative set of databases spanning multiple domains (e-commerce, Q&A platforms, medicine and sports) and a wide variety of tasks.

- The authors publicly release a pipeline for evaluation of the methods and keeping a leaderboard which will streamline adoption of the benchmark.

- The user study with an experienced data scientist provides valuable insights on the opportunities for automation in the context of predictive tasks on relational content.

**Additional Feedback:**

-

**Documentation:**

Yes. All data is sourced from publicly available repositories with licenses permitting usage for research purposes.

**Limitations:**

Yes

**Opportunities For Improvement:**

- The baselines used for entity classification and entity regression (other than the comparison against the data scientist) seem weak as they are not geared towards the RDL problem.

- The user study is limited to a single data scientist and thus the results are anecdotal in nature. While the data scientist is very experience in general, it is not clear if they have any particular expertise in the domains/tasks at hand. It would be interesting to understand what data scientists with deep expertise in those domains would produce and how that would compare against RDL.

**Relation To Prior Work:**

Yes

**Summary And Contributions:**

This paper presents a public benchmark for evaluating Relational Deep Learning (RDL) methods that consists of a broad, representative set of databases spanning multiple domains (e-commerce, Q&A platforms, medicine and sports); and a wide variety of tasks. The authors also publicly release a pipeline for evaluation of the methods and keeping a leaderboard; evaluate their previously proposed method for RDL using this benchmark; and compare against the results obtained by an experienced data scientist doing manual feature engineering (gold standard prior to RDL).

---

> ### Author Rebuttal · Authors · 2024-08-17
>
> We are grateful for your time taken to review our work. We are very glad that you have a positive assessment of our work, and that you highlight the potential impact that both the benchmark databases, and the model pipeline may have. Your review raised some points that we would like to briefly address.
>
> ---
>
> > The baselines for entity-level tasks seem weak
>
> Thanks for the good question. A key reason we do not try many graph ML methods (e.g., node2vec) is that they cannot handle temporality, and therefore are not suited to RelBench tasks.
>
> It is for this reason that we view the key baseline comparison to be the data scientist study, since this reflects the prior gold standard ML approach for relational data. The next answer below outlines why we believe that this baseline is a very strong baseline that took considerable effort to obtain.
>
> ---
> > The results of the user study are limited to a single data scientist. It would be interesting to understand how domain-specific data scientists might perform.
>
> We agree that in some cases, specialized knowledge may be useful (e.g., the clinical trials database). However, recruiting multiple data scientists was beyond our resource abilities, and would have introduced variability into the data scientist results depending on aptitudes of the different individuals. Because of this we decided to recruit just one data scientist, and optimize for the highest-skilled data scientist possible. The individual selected has significant experience in data science (5 years) and an excellent academic record during undergraduate and masters degrees at Stanford. Furthermore, the individual chosen has specific expertise in financial transitions databases, which aligns closely to at least two of our databases (Amazon and H&M).
>
> So although it is likely possible to raise the data scientist's performance further in some cases, we are confident that our user study results reflect the performance attainable by a highly-competent generalist.

---

> > ### Comment · Reviewer_Vo6o · 2024-08-24
> >
> > Thanks for your comments. I agree that the key baseline is the data scientist study. But it would be good if you could make it explicit in the paper that the data scientist study is the main baseline and all the others are secondary.
> >
> > Regarding the number of data scientists, while I still think that the study would be more significant with a larger number of data scientists, I understand the cost and effort involved in performing this task even with a single data scientist.

---

### Decision · Program_Chairs · 2024-09-26

**Decision:**

Accept (Poster)

**Comment:**

This submission generated much interest and discussion. The reviewers appreciated many aspects of the work, including (emphasis is mine) the attention to create a benchmark which uses moderate resources and can be thus run by academic labs. The back and forth during the rebuttal period helped having more solid baselines.